

# Comparison of OCO-2 target observations to MUCCnet - Is it possible to capture urban $X_{CO2}$ gradients from space?

Maximilian Rißmann[1], Jia Chen[1], Gregory Osterman[2], Florian Dietrich[1], Moritz Makowski[1], Xinxu Zhao[1], Frank Hase[3], and Matthäus Kiel[2]

[1]Environmental Sensing and Modeling, Technical University of Munich (TUM), Munich, Germany
[2]Jet Propulsion Laboratory, California Institute of Technology, Pasadena, CA, USA
[3]Institute of Meteorology and Climate Research (IMK-ASF), Karlsruhe Institute of Technology (KIT), Karlsruhe, Germany

**Correspondence:** Jia Chen (jia.chen@tum.de)

**Abstract.**

In this paper, we compare Orbiting Carbon Observatory 2 (OCO-2)'s measurements of column-averaged dry air mole fractions of $CO_2$ ($X_{CO2}$) and its urban-rural differences against ground-based remote sensing data measured by the Munich Urban Carbon Column network (MUCCnet). Since April 2020, OCO-2 regularly conducts target observations in Munich, Germany.

Its target mode data provides high resolution $X_{CO2}$ within a $15\,km \times 20\,km$ target field-of-view, that is greatly suited for carbon emission studies from space in cities and agglomerated areas. OCO-2 detects urban $X_{CO2}$ with a RMSD of less than 1 ppm when compared to the MUCCnet reference site. OCO-2 target $X_{CO2}$ is biased high against the ground-based measurements. The close proximity of MUCCnet's five fully automated remote sensing sites enables us to compare space-borne and ground-based $X_{CO2}$ in three urban areas of Munich separately (centre, north, and west), by dividing the target field into three

smaller comparison domains. Due to this more constraint collocation, we observe improved agreement between space-borne and ground-based $X_{CO2}$ in all three comparison domains.

For the first time, $X_{CO2}$ gradients within one OCO-2 target field-of-view are evaluated against ground-based measurements. We compare $X_{CO2}$ gradients in the OCO-2 target observations to gradients captured by collocated MUCCnet sites. Generally, OCO-2 detects elevated $X_{CO2}$ in the same regions as the ground-based monitoring network. More than 90% of the observed

space-borne gradients have the same orientation as the $X_{CO2}$ gradients measured by MUCCnet. During our study, urban-rural enhancements are found to be in the range of 0.1 to 1 ppm. The low urban-rural gradients of typically well below 1 ppm in Munich during our study allow us to test OCO-2's lower detection limits for intra-urban $X_{CO2}$ gradients. Urban $X_{CO2}$ gradients recorded by the OCO-2 instruments and MUCCnet are strongly correlated ($R^2 = 0.68$) with each other and have an RMSD of 0.32 ppm. A case study, which includes a comparison of one OCO-2 target overpass to WRF-GHG modeled $X_{CO2}$,

reveals a similar distribution of enhanced $CO_2$ column abundances in Munich. In this study, we address OCO-2's capability of detecting small-scale spatial $X_{CO2}$ differences within one target observation. Our results suggest OCO-2's potential of assessing anthropogenic emissions from space.





# 1 Introduction

Constantly rising atmospheric concentrations of greenhouse gases, such as carbon dioxide ($CO_2$) and methane ($CH_4$), make
combating climate change to mankind's most urgent global challenge. Even though, stringent climate targets were formulated
under the 2015 Paris Agreement aimed to limit the temperature increase to well below 2 °C, still rising anthropogenic emission
causes global mean temperatures to surge to record highs resulting in a growing number of severe and fatal weather events, that
can be linked to climate change (Shukla et al., 2019). The Annual Greenhouse Gas Index (AGGI), which is a measure for the
radiative forcing of all anthropogenic greenhouse gases (GHGs) combined, reached an all time high of 1.47 in 2021, indicating
a 47 % increase in total radiative forcing since 1990 due to rising GHG mole fractions. Especially problematic is the atmo-
spheric surge of $CO_2$, which contributes about 80 % of this growth in radiative forcing (James.H.Butler, Stephen.A.Montzka,
2021). Emissions from urban areas play a key role in this development as they are responsible for more than 70 % of global
manmade GHG emissions, even though they cover less than 3 % of land area globally (Wu et al., 2016; Gurney et al., 2015).
These numbers illustrate the importance of long term observations of $CO_2$ mole fractions, especially in large and middle sized
cities as well as closely monitoring short term $X_{CO2}$ fluxes on a sub-city scale, which gives insights on anthropogenic emis-
sion and can provide policy makers with the information needed to enact more efficient and improved emission reduction
policies. The Total Column Carbon Observing Network (TCCON) is a global network of Fourier Transform Infrared (FTIR)
spectrometers of the type Bruker IFS 125HR at 25 sites in a multitude of longitudinal and latitudinal zones (Wunch et al.,
2011). It monitors the long term atmospheric growth of $CO_2$, CO and $CH_4$ along with other atmospheric trace gases. Reg-
ular calibration against aircraft measurements make it currently to the primary validation source for other space-based $X_{CO2}$
data products (GOSAT, GOSAT-2, OCO-3, TROPOMI). Other ground-based networks like the Collaborative Carbon Column
Observing Network (COCCON) aim to improve spatial coverage by operating the low cost and portable Bruker EM27/SUN
spectrometers, which are also well suited as ground-based references for OCO-2 validation efforts (Jacobs et al., 2020; Frey
et al., 2019).

In recent years, these instruments have been used in measurement campaigns that aim to quantify urban anthropogenic emis-
sions by combining differential column measurements (DCM) and atmospheric transport data (Chen et al., 2016). Multiple
field campaigns that been carried out in Berlin (Hase et al., 2015), Munich (Dietrich et al., 2021), Indianapolis (Jones et al.,
2021), Poland (Luther et al., 2019, 2021), St. Petersburg (Makarova et al., 2021) and Hamburg (August, 2021) to show the
potential of top-down emission estimates and can help uncover unknown emission sources and constraining bottom-up inven-
tories.

In addition to the increasing number of ground-based instruments, the constantly improving space-borne remote sensing sys-
tems drastically enhance the global coverage of precise $X_{CO2}$ measurements even in hard to reach, solitary areas. NASA's
Orbiting Carbon Observatory instruments (OCO-2 and OCO-3) capture $X_{CO2}$ in four different measurements modes: nadir,
glint, target and snapshot area mode (SAM). OCO-2 captures $X_{CO2}$ on a 14-day ground-track repeat cycle (Osterman et al.,
2020). Previous studies investigated urban to rural $X_{CO2}$ enhancements (Park et al., 2021) and extracted $CO_2$ emission signals
from OCO-2 nadir tracks (Wu et al., 2018; Shekhar et al., 2020). Recently, Kiel et al. (2021) compared OCO-3 SAM and target



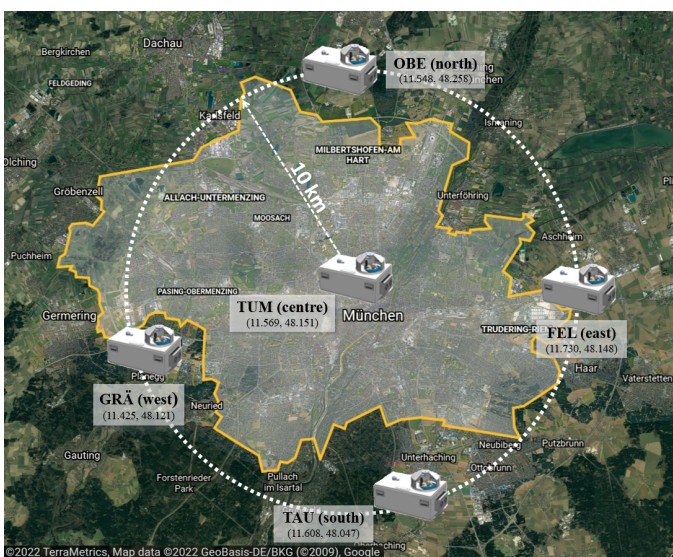

**Figure 1.** Locations of EM27/SUN spectrometers in Munich. The centre site is located on the roof of the TUM building in Munich. The other spectrometers are distributed around Munich in each compass direction.

observations over the Los Angeles Mega city against simulated $X_{CO_2}$ from two different models. This study showed that spatially fine scale satellite measurements have the potential to resolve $X_{CO_2}$ differences on a sub-city scale. Even though OCO-2 and OCO-3 measurements are evaluated against TCCON observations on a regular basis, these comparisons are performed on
a global scale and do not provide information about OCO-2's data quality on sub-city scales. In this study, for the first time, we test OCO-2's capability to determine sub-city $X_{CO_2}$ differences within one target field (approx. 15 x 20 km) by comparing OCO-2 target soundings against measurements of the Munich Urban Column Concentration network (MUCCnet). MUCCnet is a novel, fully automated ground-based network that continuously measures $CO_2$, $CH_4$ and CO column concentrations at its five sites in and around Munich (Dietrich et al., 2021). The close proximity of the ground-based instruments allows us to
compare absolute OCO-2 $X_{CO_2}$ in different parts of Munich and also lets us evaluate space-borne $X_{CO_2}$ enhancements. This way, we test the capability of OCO-2 to resolve small-scale urban $X_{CO_2}$ fluxes in Munich and other small to middle sized cities from space, which is needed to study sector dependent emissions in the future.

## 2   Datasets

### 2.1   MUCCnet $X_{CO_2}$ data

The solar spectra that are acquired by the five MUCCnet EM27/SUN devices are evaluated by two retrieval algorithms (Dietrich et al., 2021): GGG2014 (Wunch et al., 2011) and PROFFAST (Frey et al., 2019; Alberti et al., 2021). In this study, we consider the $X_{CO_2}$ outputs of the PROFFAST retrieval algorithm (Hase et al., 2004; Frey et al., 2015) that fits atmospheric $CO_2$





by scaling a priori column profiles to match the solar spectra measured by the spectrometers (Frey et al., 2019). The software is developed and maintained by the Karlsruhe Institute of Technology (KIT). The PROFFAST algorithm considers the Instrumental Line Shapes (ILS) of the individual EM27/SUN devices to reduce systematic instrument specific errors in the trace gas retrieval (Frey et al., 2015; Alberti et al., 2021). The ILS parameters, phase error (PE) and modulation efficiency (ME) of the instruments at the maximum optical path length ($OPD_{max}$) as derived from open path measurements under controlled ambient conditions (Frey et al., 2019). The instrument-specific ILS parameters as resulting from the open path calibrations are stored in the spectra generated with the PROFFAST preprocessor and are subsequently used in the trace gas analysis (Gisi et al., 2012; Sha et al., 2020).

The remaining instrument- and gas-specific discrepancies are determined by analysing side-by-side solar observations performed at the calibration facility of the COllaborative Carbon Column Observation Network (Frey et al., 2019). A reference COCCON instrument (serial number SN37) and a collocated TCCON spectrometer in Karlsruhe serve as standard of comparison. The resulting empirical corrections summarize all remaining unexplained instrument-specific corrections and are applied as instrument specific calibration factors $K_{XCO2}^{SN}$ on the raw $X_{CO2}^{raw}$ values generated by the PROFFAST retrieval code:

$$X_{CO2}^{scaled} = K_{XCO2}^{SN} \cdot X_{CO2}^{raw} \tag{1}$$

This indirectly ties the MUCCnet $X_{CO2}$ retrievals to the TCCON site in Karlsruhe since the COCCON reference device is calibrated against the TCCON site in Karlsruhe (Alberti et al., 2021; Frey and Gisi, 2021). Each MUCCnet spectrometer is protected by an enclosure, which is equipped with a multitude of sensors to fully automate the retrieval process (Heinle and Chen, 2018; Dietrich et al., 2021). Among others, the enclosures are equipped with a low cost air pressure sensor (Model 61302, Young (2009)) that captures the ground-pressure inputs for the PROFFAST retrieval. The pressure sensor of the MUCCnet centre site (TUM) is used to calibrate the other four in-situ pressure sensors. The sensors are calibrated by subtracting constant offsets which are determined performing side-by-side measurements. Pressure calibration offsets, instrument specific calibration factors and the ILS parameters are listed in Table 1.

| Serial Number (SN) | Location | Longitude (degree) | Latitude (degree) | ME | PE (rad) | $K_{CO2}$ | $\Delta p$ (hPa) |
|---|---|---|---|---|---|---|---|
| 61 | TUM_I | 11.569 | 48.151 | 0.9830 | 0.0013 | 0.9993 | 0.00 |
| 86 | FEL | 11.73 | 48.148 | 0.9830 | 0.0031 | 1.00242 | -0.2686 |
| 115 | GRÄ | 11.425 | 48.121 | 0.9837 | 0.0024 | 0.999786 | 0.0953 |
| 116 | OBE | 11.548 | 48.258 | 0.9875 | 0.0044 | 0.999973 | 0.2621 |
| 117 | TAU | 11.608 | 48.047 | 0.9791 | 0.0038 | 1.000220 | 0.4656 |

**Table 1.** Enclosure positions and EM27/SUN input parameters, that are used for calibrating PROFFAST outputs of the five MUCCnet measurement sites. (Frey and Gisi, 2021; Dietrich et al., 2021)



A global post-correction factor, that depends on the solar zenith angle (SZA), is applied to remove an erroneous low bias in the order of 0.5 ppm in the $X_{CO2}$ retrieval outputs of the current PROFFAST version (Dubravica and Hase (2021), distributed before December, 2021). The following formula removes the low bias in the scaled $X_{CO2}$ retrievals:

$$X_{CO2}^{calibrated} = \left[1.0018 - (SZA/90°)^2\right] \cdot X_{CO2}^{scaled} \tag{2}$$

We tested how this post-processing correction (see Eq. 2) impacts our $X_{CO2}$ validation results. We found that applying the post-correction to the PROFFAST retrieval effectively reduced the bias between MUCCnet and OCO-2. Hence, we can confirm that the preliminary measure is effective and should be used with PROFFAST outputs of the current software version. (Dubravica and Hase, 2021). The $X_{CO2}$ post correction will be removed in the new version of PROFFAST, which already is available to users as a beta version.

The PROFFAST retrieval and calibration process for individual devices ties the data to the COCCON network and via its connection to TCCON to the WMO trace gas scale (Frey and Gisi, 2021). All results of this paper are based on the fully scaled and calibrated retrievals $X_{CO2}^{calibrated}$.

## 2.2   OCO-2 $X_{CO2}$ data

The OCO-2 instrument was developed by NASA and launched into space on July 2nd, 2014. It orbits the earth as part of the Afternoon satellite train (A-train) at an altitude of 705 km (Crisp, 2011). Its instruments capture solar radiance spectra in one of three observational modes: nadir, glint and target mode. During OCO-2 target observations the instrument scans a certain area of interest, which is around 15 x 20 km in size. To maximize the number of soundings during one overpass the instruments scans the target area for approximately 2 minutes. The instrument captures eight spatially separated footprints simultaneously

every 1/3 of a second, theoretically yielding around 4000 measurements per overpass (Crisp, 2011). One 1.29 x 2.29 km OCO-2 footprint covers an area of just under 3 $km^2$ (Osterman et al., 2020).

The captured solar radiance spectra are processed by the Atmospheric Carbon Observations from Space (ACOS) retrieval software. In this work we use the OCO-2 lite files that are processed with the latest version (v10) of the ACOS retrieval algo-

rithm (O'Dell et al., 2018; Kiel et al., 2019). The corresponding files are publicly available and can be downloaded through the NASA Goddard Earth Science Data and Information Services Center (GES DISC, 2021). Footprint related biases and parametric biases are removed for $X_{CO2}$ retrievals in the OCO-2 lite files. A comprehensive overview of the OCO-2 and OCO-3 data products and the bias correction procedure is given in Osterman et al. (2020). Furthermore, a global scaling factor, derived from regular comparisons of OCO-2 target observations and 29 collocated TCCON sites, is applied to the $X_{CO2}$ lite file data.

This ties the OCO-2 lite $X_{CO2}$ to the standard trace scale for atmospheric $X_{CO2}$ of the World Meterology Organisation (WMO scale) (Wunch et al., 2017; Osterman et al., 2017). The most recent comparisons of fully bias corrected OCO-2 target $X_{CO2}$ and TCCON reveals a superb agreement ($rms = 0.86$ ppm, $R^2 = 0.97$) (Kiel, 2021). The data product contains a binary quality flag which flags low quality $X_{CO2}$ soundings ($qf = 1$). In the following, we solely consider good quality $X_{CO2}$ retrievals

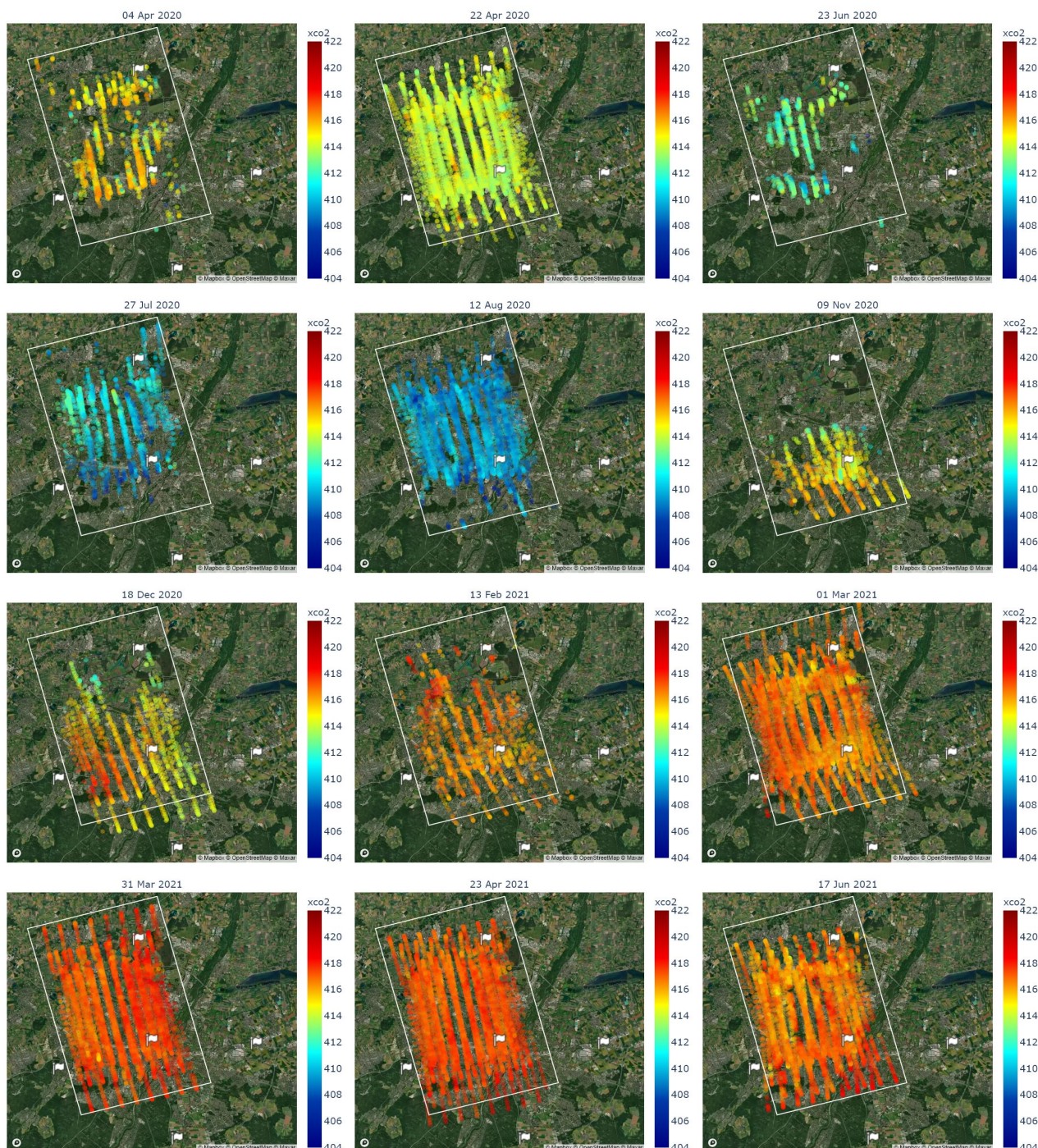

**Figure 2.** Daily $X_{CO2}$ maps of OCO-2 target observations in Munich. MUCCnet spectrometer locations are highlighted on the map.





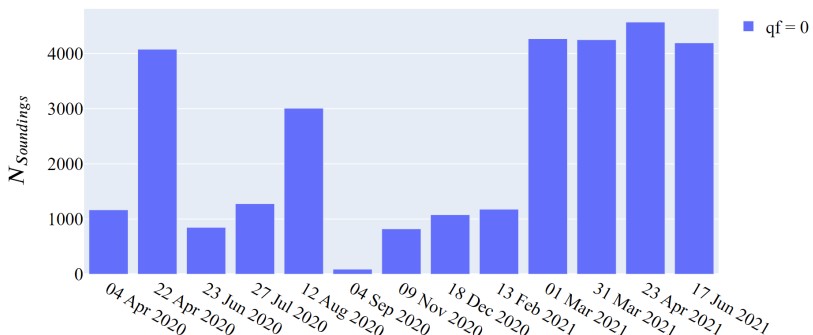

**Figure 3.** Number of soundings for each of the OCO-2 target overpasses. On most days days OCO-2 captured more than 1000 good quality ($qf = 0$) soundings per overpass. Usually a higher number of good quality soundings corresponds with more robust and less sparse data. Thus, we remove overpasses with less than 500 good quality soundings. Spectrometer locations are highlighted by flags in the target observations.

($qf = 0$) (Osterman et al., 2020).


The results of this study are based on OCO-2 target observations that took place in recent years, starting from April, 2021. From April 2020 to July 2021 OCO-2 successfully targeted MUCCnet twelve times. Figure 3 summarizes the target dates and the corresponding number of good quality ($qf = 0$) soundings. We include a target observation in our study if 1) the OCO-2 instrument gathers a minimum of 500 good quality soundings during the overpass and 2) there are ground-based retrievals for

at least one of the three sites within the target field of view. In comparison to other space-borne remotely sensed data products, the relatively small size of the OCO-2 footprints results in a higher number of good quality soundings per target observation even in cloudy conditions. One overpass, which took place on September 4th, 2020, is removed from the comparison set, since only 86 good quality $X_{CO2}$ soundings are retrieved. All remaining days had at least 800 good quality soundings. Figure 2 shows the OCO-2 $X_{CO2}$ observations of the remaining 12 *successful* overpasses over Munich. Due to OCO-2's sun-synchronous orbit,

target overpasses in Munich usually take place around 12:00 (UTC). A typical distribution of soundings is shown in Fig. 4. Three of the five MUCCnet sites are within the target field-of-view. Thus, we can compare OCO-2 $X_{CO2}$ against collocated ground-based data in the center (TUM), north (Oberschleißheim) and west (Gräfelfing) of Munich.

## 2.3 WRF model setup

We compare the OCO-2 target observations to simulated $CO_2$ column concentrations, provided by a high-resolution modeling

WRF-GHG framework designed for Munich with 45 vertical layers and a horizontal resolution of 400 m (Zhao et al., 2020b). This modeling framework is set up based on the Weather Research and Forecasting model (WRF) coupled with the biospheric flux model (Beck et al., 2011) to quantitatively understand the processes of the emission and consumption of $CO_2$ and $CH_4$ in and around Munich. The meteorological initial conditions and lateral boundary conditions in the modelled background





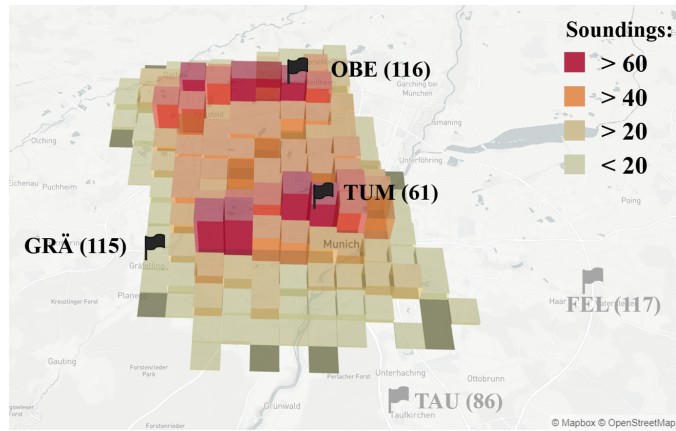

**Figure 4.** Histogram of OCO-2 target sounding distribution in Munich. There are three MUCCnet sites with sufficient collocated data, which will be considered in this study. Ground-based instruments in Feldkirchen (FEL) and Taufkirchen (TAU) are not featured in this study.

concentrations are obtained from the Integrated Forecasting System (IFS) Cycle 47r1, implemented by the European Centre

for Medium-Range Weather Forecasts (ECMWF) with a horizontal resolution of approx. 40 km. Near-surface emissions are initialized from the first version of the TNO-GHG and co-emitted species emission database (TNO_GHGco_v1.1; Super et al. (2020)). The details on the model setup and related assessment can be found in Zhao et al. (2020a). $X_{CO2}$ in the study area is derived from the modelled concentration profiles with an appropriate pressure weighing, following the method described in Zhao et al. (2019). Furthermore, WRF-GHG $XCO_2$ are binned onto an $0.01°$ x$0.01°$ latitude-longitude grid to better match the

footprint size of OCO-2.

## 3 Methods

### 3.1 Target Collocation

We use a similar methodology to Wunch et al. (2017) in order to evaluate the OCO-2 column concentrations over Munich against MUCCnet. To compare both data sets we consider the mean of all good quality OCO-2 soundings within the target

area and the ground-based $X_{CO2}$ measurements of the MUCCnet centre site $(11.569°E, 48.151°N)$, that have been recorded within $\pm30$ min of the spacecraft's overpass time. Target observations that had less than 500 good quality soundings are not considered in the comparison process. Only the target observation on September 4, 2020 does not meet this requirement.

To account for differences in the MUCCnet and OCO-2 vertical sensitivities, we apply an averaging kernel correction following the approach of Wunch et al. (2011). Hereby, the ground-based $X_{CO2}$ is smoothed with the ACOSv10 column averaging kernel

as described in Nguyen et al. (2014). We perform a York regression (York et al., 2004) to determine the best fit line and slope. (Wu and Zhen Yu, 2018).



## 3.2 By-Site Collocation

Due to the short distance of around 10 km between the MUCCnet instruments, three of the five MUCCnet sites are within the 15x20 km$^2$ OCO-2 target field-of-view. This lets us evaluate the space-borne $X_{CO2}$ retrievals for different parts of the city.

We compare subsets of OCO-2 soundings in each target observation to the $X_{CO2}$ measurements of the closest ground-based instrument.

For a collocation radius of $r_{col} = 6$ km around the spectrometer locations we achieve the highest number of collocated soundings for each site while having almost none overlap of collocated soundings between the sites (most soundings are collocated to only one MUCCnet site). This way, we section the target observation into three comparison domains - centre, west and

north. For each domain we validate space-borne measurements against $X_{CO2}$ data of the collocated MUCCnet spectrometers in Gräfelfing (west, GRÄ), Oberschleißheim (north, OBE) and Munich city centre (centre,TUM). Figure 5 shows the OCO-2 target (taken on March 31, 2021) field-of-view (white square) and the footprint positions of $X_{CO2}$ soundings. The OCO-2 soundings are colour-coded according to their comparison domain (centre = green, west = blue, north = red). The same colour-coding is used for the validation results in section 4.2.

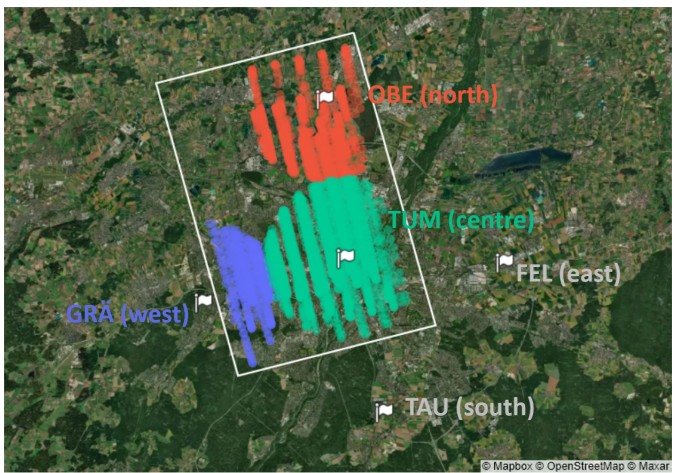

**Figure 5.** This figure illustrates the collocation criteria for target overpass data captured on March 31, 2021. OCO-2 soundings within a radius of 6 km are compared to measurements of the collocated MUCCnet instrument. The OCO-2 target soundings are coloured according to their collocated ground-based spectrometers.

The mean $X_{CO2}$ of OCO-2 soundings in each domain is compared to $X_{CO2}$ measurements of the corresponding MUCCnet site within ±30 min of overpass time. Due to the smaller size of the by-site comparison domains we only consider comparison sets, if 1) at least 70 space-borne soundings are recorded within the collocation area around each MUCCnet site and 2) the collocated ground-based instrument captured at least 50 retrievals within ±30 min of the overpass. On June 17, 2021, we extended the collocation time for the northern site in Gräfelfing to ±60 min, due to sparse ground-based measurements at



the exact time of the overpass. Figure 6 shows the number of soundings collected by the OCO-2 instruments in each domain $N_{domain}$ for the twelve target observations investigated in this paper.

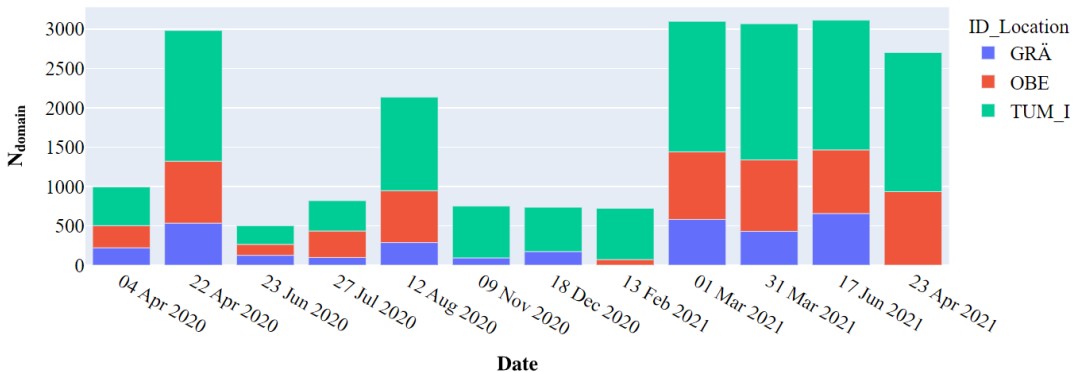

**Figure 6.** Number of good quality OCO-2 soundings $N_{domain}$ in the three comparison domains.

### 3.3 Gradient Comparison

We evaluate space-borne $X_{CO2}$ differences in the OCO-2 target field-of-view between the three seperate domains (centre, north and west) against measurements of the collocated MUCCnet spectrometers. We compute the urban gradients, present in the OCO-2 overpasses, by subtracting the mean $X_{CO2}$ of soundings collocated to one of the MUCCnet sites (domain1) from the mean $X_{CO2}$ of soundings captured in one of the other two comparison areas (domain2).

$$\Delta XCO2_{OCO-2}^{domain1-domain2} = XCO2_{OCO-2}^{domain1} - XCO2_{OCO-2}^{domain2} \tag{3}$$

This way, we compute three sets of space-borne $X_{CO2}$ gradients that are present in the target observation: 1) west-centre 2) north-centre 3) north-west. Spaceborne $X_{CO2}$ gradients are compared to the $X_{CO2}$ gradients of ground-based measurements of the collocated MUCCnet spectrometers. Ground-based gradients $\Delta XCO2_{MUCCnet}$ are computed by using $X_{CO2}$ data of the collocated MUCCnet sites.

$$\Delta XCO2_{MUCCnet}^{site1-site2} = XCO2_{MUCCnet}^{site1} - XCO2_{MUCCnet}^{site2} \tag{4}$$

Consequently, $X_{CO2}$ gradients computed between Munich centre and Gräfelfing will also be referred to as "west-centre" gradients, while those between the centre site and Oberschleißheim are called "north-centre" gradients. Positive gradients are obtained if site1 measures higher $X_{CO2}$ than site2. We use the variance of the means (standard error) to denote the error in the computed $X_{CO2}$ gradients between two domains:

$$SE_{domain1-domain2} = \sqrt{\frac{\sigma_{domain1}^2}{N_{domain1}} + \frac{\sigma_{domain2}^2}{N_{domain2}}} \tag{5}$$

When compared to the by-site comparison process (Sec. 4.2), we apply stricter criteria to filter which overpasses are considered to be robust and suited for the gradient assessment. We exclude space-borne $X_{CO2}$ gradients, if the mean space-borne $X_{CO2}$





in one of the domains is computed using less than $N_{domain} = 100$ soundings and if it has a standard deviation larger than
$\sigma = 0.75$ ppm. This criteria removes two of the gradients from the set (on July 27, 2020 and November 9, 2020). Second, we
checked MODIS images taken at overpass time for high cloud coverage. On June 23, the MODIS images and a high aerosol
contamination point at challenging measurement conditions causing a sparse distribution of converged soundings around the
MUCCnet centre site. Therefore, we do not consider urban $X_{CO2}$ gradients captured on June 23, 2020.

## 4   Results

### 4.1   OCO-2 target validation

To test the agreement of OCO-2 and MUCCnet $X_{CO2}$, we perform a york regression between the twelve OCO-2 target obser-
vations and the $X_{CO2}$ measurements of the MUCCnet reference instrument in the centre of the OCO-2 target field-of-view. The
results are shown in Fig. 7. For all target observations that are considered in this study, the root mean square $X_{CO2}$ difference is
below 1 ppm ($RMSD = 0.96$ ppm). Furthermore, the coefficient of determination $R^2 = 0.93$ reveals a very strong correlation.
On average, the space-borne $X_{CO2}$ is about 0.70 ppm higher than the collocated solar measurements taken by the MUCCnet
reference device. This high bias is comparable to the observed bias when OCO-2 target data is compared to the Karlsruhe TC-
CON instrument (bias= 0.80 ppm, RMSD=0.91 ) to which MUCCnet is tied (as discussed in Sec. 2.1). The RMSD improves to
0.66 ppm when the bias between the space and ground-based measurements is not taken into account (Matthaeus Kiel, email
correspondence, January 26, 2022). The averaging kernel correction that we applied to the $X_{CO2}$ data improves the root mean
square difference by 0.18 ppm. Similar effects of the averaging kernel correction are also observed in Kiel et al. (2021).

Figure 8 shows the differences between space-borne and ground-based $X_{CO2}$ retrievals for each individual overpass. Both
observing systems observe a similar seasonal behaviour of urban $X_{CO2}$ in Munich in the time period analyzed here. Both
OCO-2 and MUCCnet measurements also capture a similar seasonal cycle in $X_{CO2}$. For eleven out of the twelve overpasses,
OCO-2 measured the higher concentrations, causing an average OCO-2 high bias of 0.7 ppm. Only one overpass (August 12,
2020), OCO-2 captured lower mean $X_{CO2}$ than MUCCnet. On two days a mean offset higher than 1.5 ppm is measured, which
is likely caused by sub-optimal measurement conditions. On November 9, 2020 more than 80 % of retrievals are low quality
($qf = 1$) yielding just 816 usable soundings. In the study time, the bias in the satellite data does not show a noticeable temporal
drift. The four overpasses in which the space-borne $X_{CO2}$ offset deviates the most from the mean bias took place between July
27, 2020 and December 18, 2020.

### 4.2   By-site validation

Dividing the OCO-2 target observations into spatially separated comparison domains allows us to validate the space-borne
$X_{CO2}$ in the north (OBE), center (TUM) and west (GRÄ) of Munich. In each domain we consider space-borne soundings





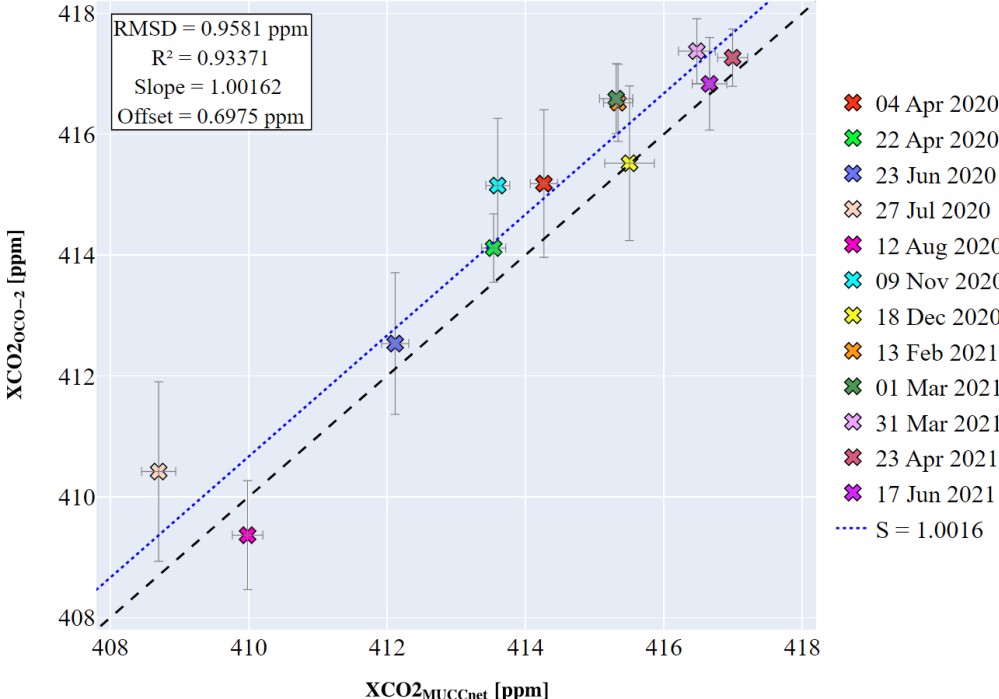

**Figure 7.** Scatter plot of MUCCnet and OCO-2 $X_{CO2}$ in the comparison time frame. Each overpass is color-coded.

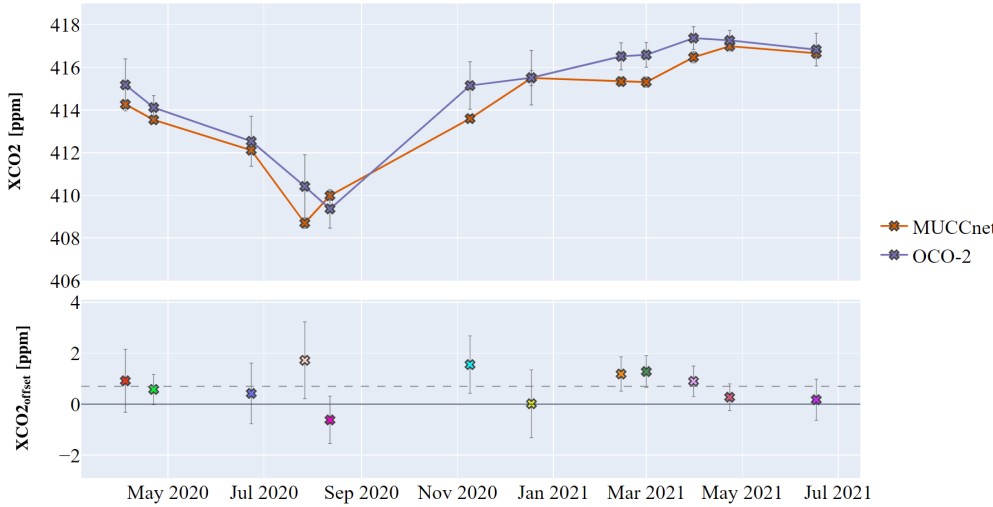

**Figure 8.** Both observing systems detect the seasonal $X_{CO2}$ variations and a $X_{CO2}$ growth in the study period. The lower panel shows the daily $X_{CO2}$ differences of satellite data and MUCCnet observations. The space-borne observations are biased high by 0.7 ppm.





that are within a 6 km distance of the collocated MUCCnet instrument. The centre domain usually has the highest amount of space-borne soundings, because the northern and western instruments are closer to the edge of the target area. In contrast to the target comparison in section 4.1 we consider a spatially more constrained subset of OCO-2 soundings. This improves the root mean square differences of OCO-2 and the MUCCnet centre site to $RMSD_{TUM} = 0.82$ ppm. This improvement

is caused due to the more specific collocation, that reduces the effect of averaging over potential spatial $X_{CO2}$ gradients in the OCO-2 target observation. The scatter plots in Fig. 9 show the by-site comparison results for target observations in the study. Similar to the results in the centre domain, for the two remaining MUCCnet sites, we find RMSD values of less than 1 ppm ($RMSD_{GRÄ} = 0.61$ ppm and $RMSD_{OBE} = 0.94$ ppm) when compared to the collocated ground-based measurement sites. Furthermore, all three scatter plots show improved coefficients of determination ($R^2_{TUM} = 0.96$, $R^2_{GRÄ} = 0.97$ and

$R^2_{OBE} = 0.96$) when compared to the target validation results in Section 4.1.

We computed a high bias of OCO-2 against the MUCCnet spectrometers in all three comparison domains ranging from $b_{GRÄ} = 0.36$ ppm over $b_{TUM} = 0.59$ ppm to $b_{OBE} = 0.78$ ppm. We assume, the relatively small sample size of eleven overpass days and measurement uncertainties to cause the discrepancies in the computed biases. The differences in the relative location of the collocated OCO-2 soundings in the target field-of-view could impact the results due to changes in the

viewing geometry of the space-borne instruments. A larger sample size is required to make a more robust statement. The best fit RMSE is nearly identical for all three comparison domains ($RMSE_{TUM} = 0.57$ ppm, $RMSE_{GRÄ} = 0.57$ ppm and $RMSE_{OBE} = 0.57$ ppm).. A summary of the linear regression results for target and by-site validation is given in Table 2.

| Domain | RMSD (ppm) | $R^2$ | Bias (ppm) | RMSE (ppm) |
|---|---|---|---|---|
| centre (TUM) | 0.822 | 0.957 | 0.594 | 0.57 |
| west (GRÄ) | 0.608 | 0.973 | 0.360 | 0.57 |
| north (OBE) | 0.939 | 0.963 | 0.776 | 0.57 |
| Target Comparison (Sec. 4.1) | 0.958 | 0.934 | 0.698 | 0.66 |

**Table 2.** Regression results of collocated $X_{CO2}$ measured by MUCCnet and OCO-2 of by-site and target validation. In all domains OCO-2 is biased high against MUCCnet.

The daily offsets in each domain are depicted in Fig. 10. OCO-2 measures higher $CO_2$ mole fractions than MUCCnet in all three domains, each overpass except for August 12, 2020. For most overpasses the by-site offsets are more or less consistent in

each of the three collocation areas. The largest discrepancies in daily offsets in the three domains could be observed on overpass days with a smaller than average number of good quality soundings (e.g. November 9, 2020 and July 27, 2020). Target observations with a high number of good quality soundings in general have had smaller differences in daily by-site $X_{CO2}$ offsets.

Overall, we discover an improvement of RMSD and a higher correlation $R^2$ in the by-site validation when compared to the

target validation due to the smaller collocation radius. OCO-2 is well capable of detecting $X_{CO2}$ in the three domains in the centre, west and north of Munich. However, small differences in averaged bias are present in the three collocation areas.





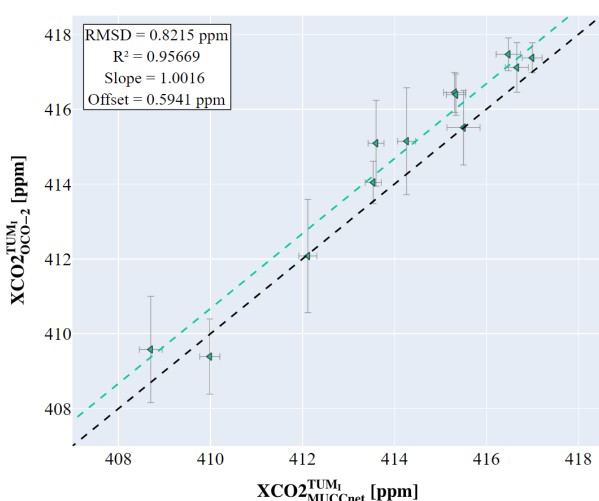

(a) MUCCnet centre $X_{CO2}$ versus collocated OCO-2 $X_{CO2}$ (centre domain)

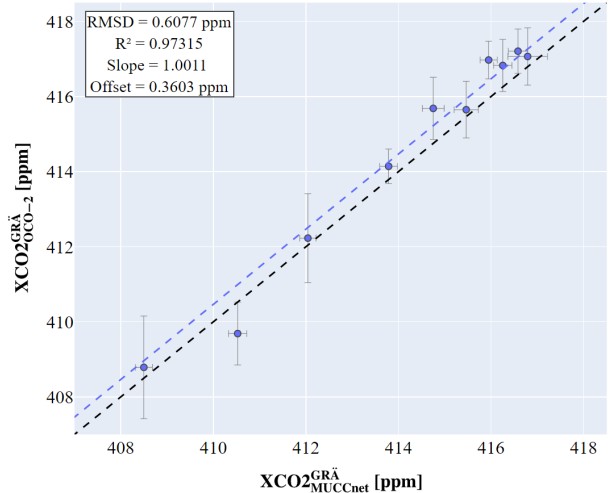

(b) MUCCnet $X_{CO2}$ in Gräfelfing versus collocated OCO-2 $X_{CO2}$ (western domain)

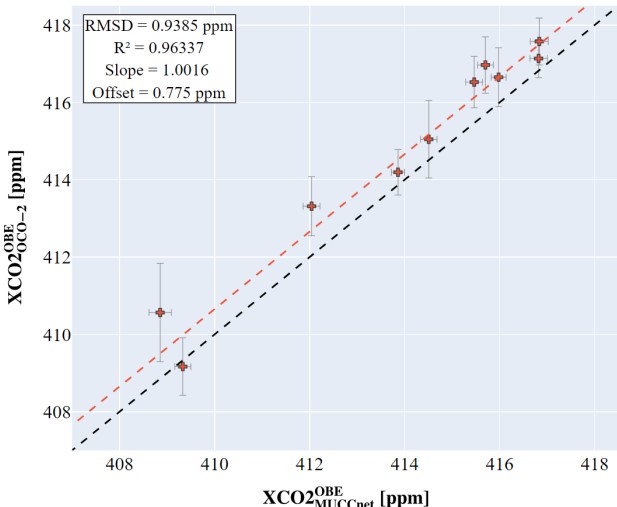

(c) MUCCnet $X_{CO2}$ in Oberschleißheim versus collocated OCO-2 $X_{CO2}$ (northern domain)

**Figure 9.** By-site comparison results between OCO-2 and MUCCnet. We use the same color coding as in Fig. 5. OCO-2 has the largest bias and RMSD in the northern domain.

## 4.3 Assessment of urban $X_{CO2}$ gradients measured from space

The adjusted collocation procedure also allows us to assess the space-borne $CO_2$ gradients in the OCO-2 target observations. This is the first time gradients within one OCO-2 target observations are evaluated against measurements of multiple ground-based measurements sites. We contrast the space-borne $\Delta XCO2_{OCO-2}$ against the $X_{CO2}$ differences measured by the





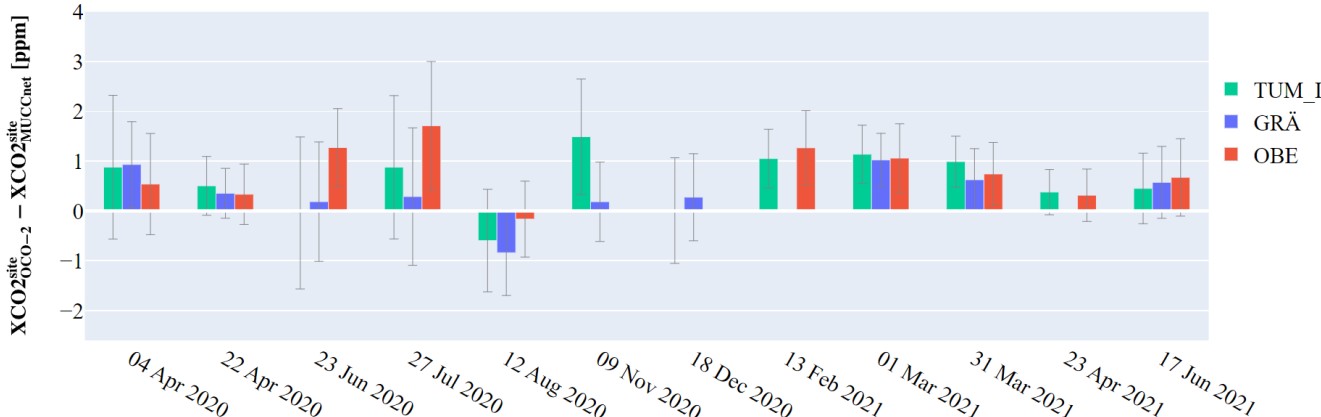

**Figure 10.** Daily offsets of collocated $X_{CO2}$ captured by OCO-2 and MUCCnet in each comparison domain. During most overpasses the by-site offsets are alike for all comparison domains. However, during some overpasses (eg. Jun 23, Jul 27, Nov 9) we observe a higher level of intra-day variation of the daily by-site offsets, which can impede the detection of urban gradients.

collocated MUCCnet spectrometers during the overpass (see Fig. 11). This simple approach allows us to see how spatial $X_{CO2}$ gradients in the target observations compare to the $X_{CO2}$ differences captured by MUCCnet. We compute three sets of gradients (north-west, north-centre and west-centre), for each overpass, where a sufficient amount of data is available.

$X_{CO2}$ enhancements in Munich are usually in the range from 0.1 to 1.0 ppm during the overpasses featured in this study. This coincides with results of previous urban gradient assessments in Munich published in Dietrich et al. (2021). On average, the MUCCnet instruments measured site to site enhancements of 0.42 ppm. These rather low gradients allow us to test the lower detection limits of OCO-2 for resolving small scale gradients.

Considering the rather small $X_{CO2}$ gradients in Munich, OCO-2 detects the elevated $X_{CO2}$ in the same domain as MUCCnet for 20 of the 22 computed gradients and therefore qualitatively determines the area of enhanced $X_{CO2}$ correctly in 91 % of cases. Furthermore, for 68 % (15/22) of the computed gradients, OCO-2 is within a margin of error of just 0.25 ppm when compared to the more precise MUCCnet measurements. For the entire set of gradients OCO-2 achieved an $RMSD$ of 0.31 ppm and a linear correlation with a strong correlation of $R^2 = 0.68$ between OCO-2 and the MUCCnet measurements.

In particular, for west-centre gradients (between TUM and Gräfelfing) space-borne and ground-based $X_{CO2}$ gradients are highly correlated ($R^2_{west-centre} = 0.89$) with a $RMSD = 0.21$ ppm. The space-borne north-west and north-centre gradients have higher RMSDs and are moderately correlated ($R^2_{north-west} = 0.39$, $R^2_{north-centre} = 0.54$) to the $X_{CO2}$ gradients measured by the MUCCnet spectrometers. For the north-west and north-centre $X_{CO2}$ differences the RMSD is 0.33 ppm and 0.36 ppm, respectively. Due to the low sample size, the space-borne $X_{CO2}$ gradients captured on July 27, 2020 and August 12, 2020

strongly impact the regression results for the north-west and north-centre subsets. Here, north-west and north-centre $X_{CO2}$



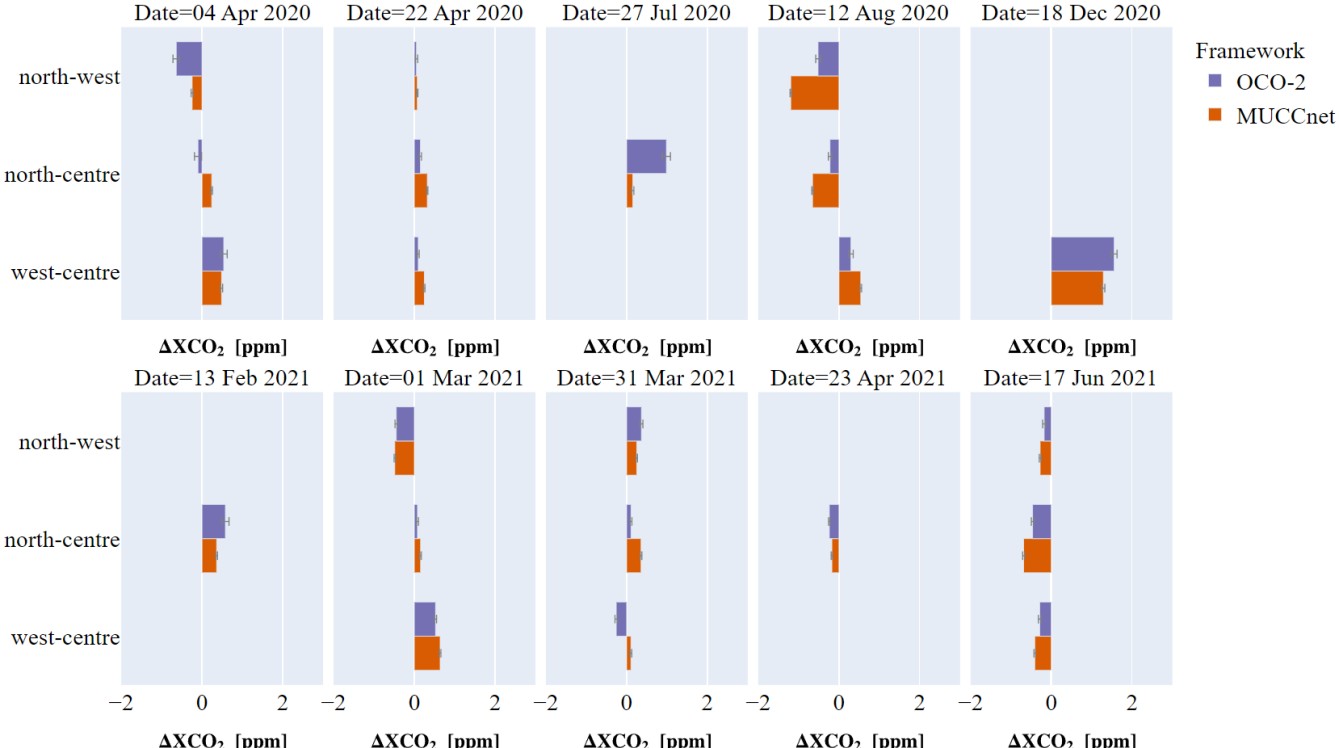

**Figure 11.** $X_{CO2}$ gradients in Munich on overpass days. Blue bars represent the gradients present in the OCO-2 target observations. Orange bars denote $X_{CO2}$ gradients captured by MUCCnet. On most days, OCO-2 sees elevated $X_{CO2}$ in the same region as the ground-based MUCCnet instruments. Error bars denote the standard error of the two means (see Eq. 5)

differences, captured by OCO-2, are off more than 0.5 ppm. During both overpasses, we observe a higher $X_{CO2}$ offset in the northern domain than in the other two domains (see Fig. 10). Due to the low absolute $X_{CO2}$ gradients that are captured during our study and the relatively low sample size, single outlier overpasses have a strong impact on the regression results. Conse-
quently, if we remove both outlier days, July 27 and August 12, we achieve an overall improved RMSD and strong correlation for all subsets of gradients. These improved results are shows in Fig. A2 in the Appendix A2. We expect more robust and definitive results for a larger sample size. It's important to be aware of the measurement context. Generally, we see a better agreement in gradients for days with a high yield of good quality soundings and good measurement conditions.

There is no tendency towards one observational method showing systematically higher or lower gradients than the other. On
some days, MUCCnet measured greater $X_{CO2}$ enhancements in the suburban sites when compared to the OCO-2 observations like it is the case on April 22, 2020 and March 1, 2021. During the overpasses on December 18, 2020 and February 2, 2021, OCO-2 detected slightly higher $X_{CO2}$ gradients than MUCCnet.





(a) All three subsets of $X_{CO2}$ gradients show a strong correlation.

(b) North-west $X_{CO2}$ gradients.

(c) West-centre $X_{CO2}$ gradients

(d) North-centre $X_{CO2}$ gradients

**Figure 12.** Linear regression results of space-borne and ground-based $X_{CO2}$ differences. Depending on the subset of gradients we observe moderate to very strong correlation between ground-based and space-borne gradients. These differences in agreement are caused by single outliers, which impact the regression results due to the small sample size and low absolute gradients in Munich.

The overall strong correlation shows that OCO-2 is capable of detecting similar mean $X_{CO2}$ differences as MUCCnet. Even though the spread of the space-borne measurements in each domain is sometimes larger than the gradients itself, the $X_{CO2}$ means in each domain are robust enough to capture the small urban gradients between the domains from space. These results show that OCO-2 target observations capture valuable information about the spatial distribution of $X_{CO2}$ within one OCO-2 target field-of-view. If the measurement conditions are good, OCO-2 target mode can successfully capture urban $X_{CO2}$ gradients





in Munich. It leads to the conclusion that OCO-2 is capable of detecting intra-urban $X_{CO2}$ fluxes and enhancements, caused by

anthropogenic activities on a sub-city scale. Hence, OCO-2 target observations could find more use in assessing area sources of $CO_2$ from space.

## 4.4    $X_{CO2}$ enhancements on December 18, 2020

December 18, 2020 was the only overpass on which ground-based centre-west $X_{CO2}$ enhancements are greater than 1 ppm.

During the one hour overpass collocation time, $CO_2$ retrievals in Gräfelfing ($XCO2_{MUCCnet} = 416.8 \pm 0.43$) exceeded the mean $X_{CO2}$ in Munichs city center by $\Delta XCO2_{MUCCnet} = 1.3$ ppm. The center spectrometer measured $XCO2_{MUCCnet} = 415.5 \pm 0.36$ ppm during the overpass. On this day the collocated OCO-2 is in good agreement with its ground based counterpart (see Fig. 13). Hence, OCO-2 observes similarly large enhancements of $\Delta XCO2_{OCO-2} = 1.55$ ppm in the west of Munich. $X_{CO2}$ enhancements between an upwind and a downwind measurement site is caused by natural and anthropogenic emissions

and the subsequent atmospheric transport (Chen et al., 2016). We use ERA5 wind data within $\pm 2$ h of the overpass time to evaluate which of the measurement sites are positioned downwind and upwind during the overpass. On December 18, 2020 the mostly east and east-south-east winds with relatively low wind speeds of less than 1.91 m/s are reported. Thus, convective transport of anthropogenic $CO_2$ emissions in the urban centre of Munich towards the west causes enhanced $X_{CO2}$ in the western comparison domain. Both, ground-based and satellite measurements capture similar $X_{CO2}$ enhancements, that are higher than

usual.

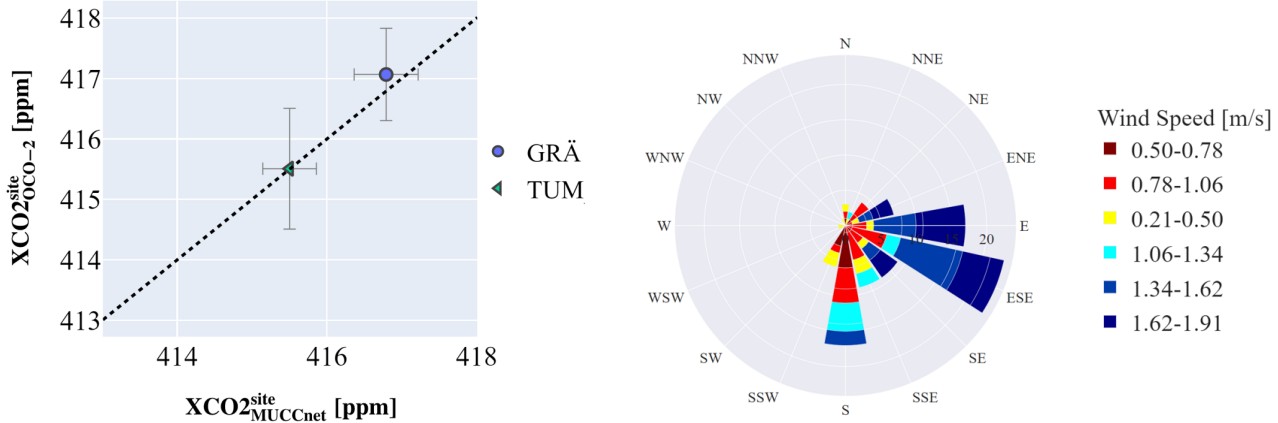

**Figure 13.** $X_{CO2}$ captured by OCO-2 and MUCCnet on December 18, 2020

**Figure 14.** ERA5 wind rose $\pm 2$ h of overpass time.

The spatial distribution of $X_{CO2}$ in Munich is shown in Fig. 15a. The lowest $X_{CO2}$ is measured in the south-east and north of Munich with increasingly higher concentrations in the centre. The highest $X_{CO2}$ is captured right at the western edge of the tar-





get field-of-view close to the MUCCnet site in Gräfelfing. Here, single soundings reach peak concentrations of up to 418.6 ppm.

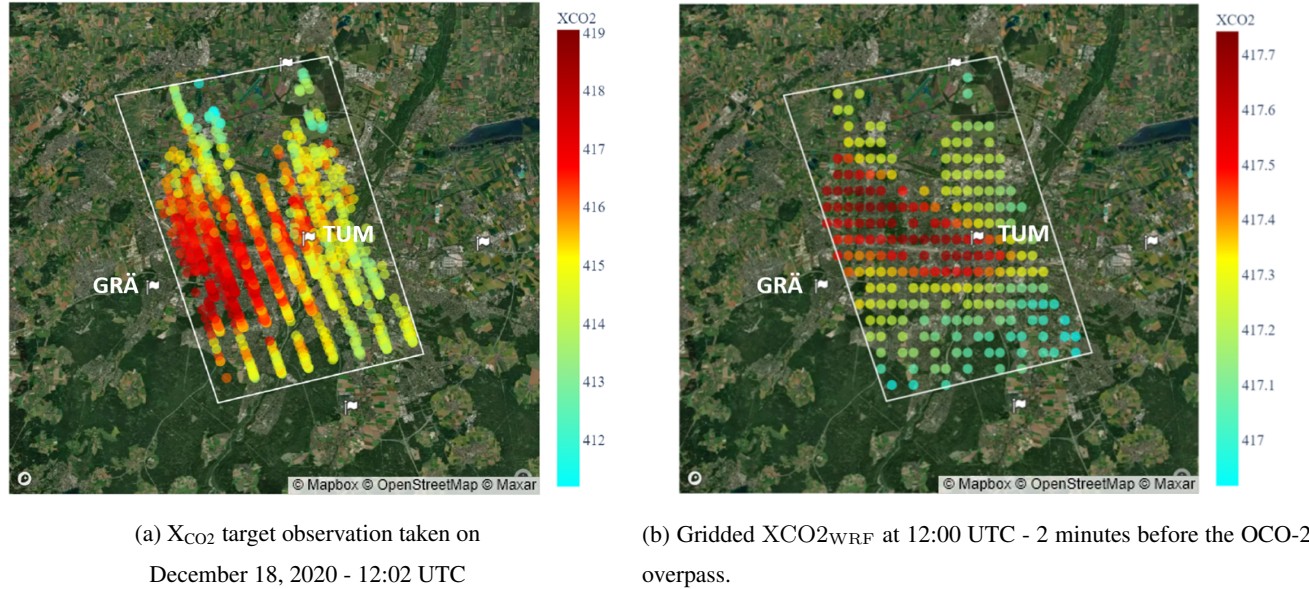

(a) $X_{CO2}$ target observation taken on December 18, 2020 - 12:02 UTC

(b) Gridded $XCO2_{WRF}$ at 12:00 UTC - 2 minutes before the OCO-2 overpass.

**Figure 15.** Gridded WRF-GHG outputs and OCO-2 target observation for December 18th, 2020. OCO-2 target field-of-view is visualized by a white square. Enhanced $X_{CO2}$ is predominantly captured in the centre west of Munich.

A qualitative comparisons of the OCO-2 target overpass to the satellite retrievals shows, that OCO-2 and the WRF-GHG produce a similar spatial distribution of urban $X_{CO2}$ during the overpass. The plots in Fig. 15 show both the $X_{CO2}$ captured by OCO-2 (left) and the $X_{CO2}$ generated via the WRF model (right). The gridded WRF results have an overall higher mean $X_{CO2}$ of $XCO2_{WRF} = 417.32 \pm 0.21$ ppm while the satellite measures $XCO2_{OCO2} = 415.152 \pm 1.3$ ppm. Nonetheless, both approaches have the highest $CO_2$ mixing ratios in the west. A plume-like shape originating in the centre of Munich extends westwards.

OCO-2 captures a broader spread of $X_{CO2}$ in contrast to the more distinct plume shape generated by the WRF-GHG model. Lowest mixing ratios are modelled and measured in the south-east and north-east. The spread of mole fractions, that is captured by OCO-2, is considerably higher than the outputs of the WRF-GHG simulation. To compare the $X_{CO2}$ differences measured by OCO-2 and MUCCnet to the WRF-GHG simulation, we compute the mean $X_{CO2}$ of the WRF-footprints within the three comparison domains (centre, west and north). Afterwards, the gradients are calculated with the same approach that is used

to compute the OCO-2 gradients. The enhancements measured ($\Delta XCO2_{OCO-2}^{west} = 1.55$ ppm and $\Delta XCO2_{MUCCnet}^{west} = 1.23$ ppm) are significantly higher than the $\Delta XCO2_{WRF}^{west} = 0.15 \pm 0.15$ ppm the forward model produces. The satellite observations resemble the precise retrievals, measured by the MUCCnet instruments, better than the WRF-GHG model. We assume this underestimation of $X_{CO2}$ gradients to be caused by both, uncertainties in the annual emission inventory as well as transport uncertainties. A mismatch in model wind speed and direction causes the area of maximum $X_{CO2}$ enhancements to be shifted

to the north in the modeled $X_{CO2}$ (see Fig. 15). Furthermore, the $X_{CO2}$ in the target observation is notably higher than on other





days, indicating unusually high emissions in Munich on December 18, 2020, which can't be replicated by an yearly-averaged bottom-up emission inventory, while the spatial distribution is reproduced rather accurately. We recognize the potential of space-borne $X_{CO2}$ retrievals in reducing the mentioned uncertainties in the model transport and emission inventories. These results suggest that for good measurement conditions and high gradients, OCO-2 target measurements can be utilized for an
accurate assessment of urban $X_{CO2}$ and its spatial distribution in middle to larger sized cities.

## 5 Conclusion

Comparisons between OCO-2 target measurements over Munich, Germany and ground-based measurements performed by MUCCnet's reference instrument agree well for the analyzed time period with a RMSD value of 0.96 ppm. On all days, OCO-2 appears to be biased high with a mean offset of 0.7 ppm. This bias is similar to comparisons between OCO-2 and the TCCON
site in Karlsruhe. In the by-site comparison we find a improved correlation and reduced RMSDs in all three spatially separated comparison domains (centre, west, north), due to the smaller collocation area, which reduces the impact of potential spatial $X_{CO2}$ gradients in the target field on the validation results.

For the first time, sub-city scale $X_{CO2}$ variations in the OCO-2 target measurements were cross compared against collocated ground-based $X_{CO2}$ gradients captured by multiple MUCCnet sites. Due to the relatively small spatial $X_{CO2}$ differences of
mostly below 0.5 ppm in Munich we were able to test the lower detection limits for sub-city scale gradients. Even though, OCO-2's proclaimed precision of 1 ppm is larger than most gradients we captured during our study, we found moderate to strong aggreement between MUCCnet and OCO-2 $X_{CO2}$ gradients and root mean square values of 0.21 to 0.36 ppm. For more than 90% of the captured gradients, OCO-2 was able to detect the correct direction of the $X_{CO2}$ gradients. The overall low $X_{CO2}$ differences in Munich and the limited number of overpasses featured in this study make it hard to draw more definitive
conclusions for now. We expect urban monitoring networks like MUCCnet to play a crucial role in validating space-borne $X_{CO2}$ gradients of wide swath $CO_2$ monitoring missions in the future. It will be interesting to see how OCO-2 and OCO-3 will perform against similar setups in megacities and larger industrial areas.

Finally, the qualitative comparisons to WRF-model data on December 18, 2020 reveal a matching spatial distribution of target and model $X_{CO2}$. Emissions in the city centre are transported westwards and cause enhanced $X_{CO2}$ close to the west-
ern MUCCnet site in Gräfelfing. This points to OCO-2's potential of locating highly potent emission sources and providing valuable insight for future model development.

All things considered, we see the potential of OCO-2 to provide vital information about urban gradients in cities, agglomerated areas and other potent $CO_2$ emitters around the globe that further improves the understanding of the relevance of anthropogenic urban emissions for our climate. We hope, the measurement of urban $X_{CO2}$ gradients from space will be a powerful
tool for evaluating urban anthropogenic emission, globally. Further comparisons of OCO-2 target observations to ground-based monitoring networks are beneficial to better understand OCO-2's capability of assessing point and area sources from space.



*Acknowledgements.* We thank Friedrich Klappenbach for his help on setting up the PROFFAST retrieval locally for the five EM27/SUN instruments, Franziska Dobler for helping with the WRF-GHG data processing. Special thanks to our colleagues at the Jet Propulsion Laboratory (JPL) and their OCO-2 team for regularly targeting Munich as well as always being supportive and providing researchers with high quality data and support.

*Author contributions.* MR wrote the manuscript in cooperation with JC. JC, MK, XZ, FD, GO and FH worked on the manuscript. FD and JC set up the ground-based remote sensing network in Munich (MUCCnet), that provides the EM27/SUN datasets. MK and GO guided the research with their expertise on the OCO ACOSv10 datasets. MK provided up to date TCCON comparison results. XZ set up the WRF-model framework in Munich and provided us with the WRF-GHG datasets. MM set up and automated the PROFFAST retrieval for all five measurement sites in Munich. MR conducted collocation and validation data processing and visualization of our results. FH provided us valuable information about the PROFFAST retrieval and post-correction. JC, MK and FD supervised the the research.

*Competing interests.* We declare no conflicts of interest with this research.

*Code and data availability.* All OCO-2 data files are available at the GES DISC data centre (GES DISC, 2021). All PROFFAST retrieval files and WRF-GHG outputs are stored locally on the ESM cloud server and are available at request. Plots are generated using the python plotly library.





## A1    Appendix I

Figure A1 shows the daily mean $X_{CO2}$ measured by OCO-2 in each comparison domain versus the collocated MUCCnet instrument. In target observations taken on June 23rd, and July 27th, 2020 strong site-to-site $X_{CO2}$ differences are visible in the OCO-2 data while MUCCnet measures little to none $X_{CO2}$ differences between its sites. On August 12th and November 9th, the opposite is true and MUCCnet measures higher $X_{CO2}$ enhancements than OCO-2.

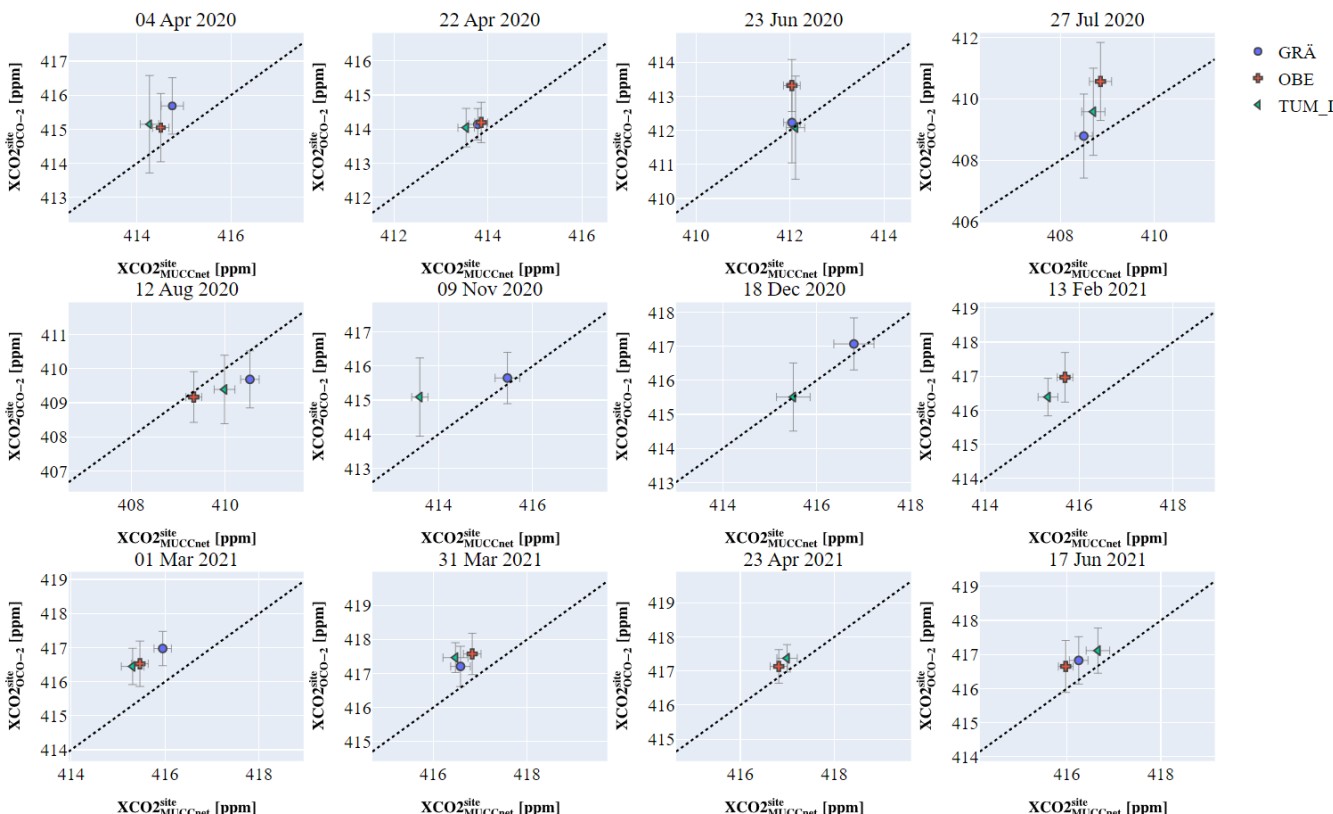

**Figure A1.** Daily $X_{CO2}$ by-site comparison results.





## A2 Appendix II

(a) All three subsets of $X_{CO2}$ gradients show a strong correlation.

(b) North-west $X_{CO2}$ gradients.

(c) West-centre $X_{CO2}$ gradients

(d) North-centre $X_{CO2}$ gradients

**Figure A2.** Linear regression results of space-borne and ground-based $X_{CO2}$ enhancements when gradients captured on Aug 18, 2020 and June 27, 2020 are removed from the set. When removing overpasses with a high difference in daily by-site offsets, which impedes the correct detection of urban gradients, we obtain a strong to very strong correlation between OCO-2 and MUCCnet gradients.



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
