# Peer review of "Comparison of OCO-2 target observations to MUCCnet - Is it possible to capture urban $X_{CO2}$ gradients from space?"

_Atmospheric Measurement Techniques, 2022_

## Referee Comment (RC1)

Review of
**Comparison of OCO-2 target observations to MUCCnet - Is it possible to capture urban
$X_{CO2}$ gradients from space?**
by Rissmann et al, 2022, submitted to AMT.

This paper compares small-spatial scale (<20 km) XCO2 gradients in OCO2 target-mode data
with those from ground-based EM27 sensors located in Munich, Germany. Though nadir mode
observations have analyzed some gradients, this is the first time multiple ground-based sensors
have been used to evaluated OCO-2's ability to measure very small-scale XCO2 gradients, in
particular from target-mode observations. This is important because it adds to the growing
evidence that OCO-2 data does accurately capture small-scale XCO2 gradients, and therefore
can be used to infer local-scale fluxes in CO2 from, e.g., urban areas and point sources (such as
power plants).

This paper is very well-written and logically laid out. I have only some minor comments that
should be addressed before publication. Once published, it will be an important addition to the
literature on the topic of CO2 measurements from space.

My first major comment involves error bars. The error bars plotted on OCO2 biases (relative to
MUCCnet), such as shown in Figures 7 & 8, appear to be standard deviation only. *IF* one
were to assume that these errors were randomly distributed over the small area (<10 km) over
which they are evaluated, we could estimate the standard error of the mean as sigma/sqrt(N),
where N is the number of observations. However, it is well known that OCO-2 errors are NOT
randomly distributed, over small or large areas (see e.g. Kulawik et al., 2019). It is likely that the
local scale mean of OCO-2 includes systematic errors that cannot easily be evaluated. This point
should clearly be made. Therefore, it is difficult to evaluate errors on mean OCO-2 values over
the whole domain or the 3 sub-domains. Further, it is equally difficult to state them for the
gradients. On this note, the error treatment on the gradients is both incorrect (as it assumes
1/sqrt(N) Gaussian averaging) and inconsistent with the stated error bars on XCO2 overpass
means (which simply uses the standard deviation of the XCO2 values in a given domain). The
current error bars listed on the gradients are unrealistically small (of order or less than 0.1 ppm).
Therefore, please expand your discussion of errors to include these points, and be sure to treat
errors consistent on the mean quantities and the gradient quantities.

Second comment (albeit minor) is how the data are presented on a map. Figure 2 shows this for
the 12 dates analyzed. I suggest averaging the data onto a 0.5x0.5 km$^2$ grid, because as you show
it now, it is large circles that overlap each other. Your plotting method emphasizes the noise in
the data, rather than the spatial gradients. It further ignores that rather large horizontal extent of
the OCO-2 FOVs, which are parallelograms and in some of these cases are rather wide! You can
see an example of this in the figure below, which shows your plot of the Dec 18, 2020 case on
the left, the corresponding NASA Worldview plot in the middle (obtainable at
https://go.nasa.gov/38MQLRr), and my own home-made generated plot on the right. You can
see the two on the right de-emphasize the noise and emphasize the gradients, and further do a
better job illustrating the fact that OCO-2 target observations provide relatively complete spatial
coverage due to the large degree of overlap of the various soundings in the image. The central
image also illustrates the value of including the actual Aqua-MODIS data, at the expense of the

spatial resolution of the surface imagery (unfortunately). But often there is cloud contamination in these Munich targets, so it is something to consider when plotting. You may wish to include something like the central image in your Figure 15, to really showcase the cloud context. And in general, please consider using one of these more realistic plotting methods which includes the actual parallelogram-shaped FOVs along with some kind of averaging.

[Figure]

Fig 15(a) from this manuscript

NASA Worldview image, with colorbar ranging from 411 to 419 ppm. There is spatial averaging of the actual OCO-2 paraollelogram-shaped FOVs. The Aqua-MODIS true color image from the same orbit is shown as the background.

IDL-generated map by this reviewer, where the paraollelogram-shaped FOVs are averaged where they overlap, to a 0.006° x 0.006° (0.67x0.45 km²) lat-lon grid.

Specific comments:

Section 3.2: The value of 6km radius around each EM27 seems rather arbitrary. Yes, it maximizes the circles while minimizing their overlap, but does it make sense as to the column of air the various MUCCnet sites are really sampling, as compared to OCO-2? For instance, using +/- 30 minutes overpass time for collocation, we could estimate that corresponds to a spatial extent of roughly +/- 15 km for a wind speed of 8 m/s, meaning that the EM27 signals will be much more "washed out" in the case of higher wind speeds, due to averaging all the EM27 values in that 1 hour time window. You should at least speak to this source of collocation error, and you may wish to mention that more sophisticated schemes (such as using a tighter time window) may reduce the collocation error.

Near line 150: Please state the spatial resolution of the anthropogenic emissions in TNO-GHG? It appears to be roughly 1x1 km^2, but if it is larger, than could also explain the Dec 18, 2020 discrepancy.

Near line 335: Please state the WRF-GHG wind speeds (roughly) in this case. If they are significantly larger than the 0.5-2 m/s wind speeds ERA5 shows, that could explain the ~factor of 10 discrepancy.

Near line 365: You ignore the factor of 10 discrepancy in the spatial XCO2 gradients between WRF-GHG and the observations. You should mention this as well, as the obs provide a potential way to improve whatever is going on in the model. I'm guessing perhaps there is something fundamentally wrong about how the model was set up and run, because I think WRF, at sufficiently high spatial resolution, should be able to duplicate the rough magnitude of observed spatial gradients.

---

## Referee Comment (RC2)

**Review of "Comparison of OCO-2 target observations to MUCCnet – Is it possible to capture urban XCO2 gradients from space?" by Rißmann et al.**

The manuscript compares target viewing mode retrievals of total-column CO2 from NASA's Orbiting Carbon Observatory version 2 (OCO-2) to ground-based portable Fourier Transform Infrared Spectrometers (EM27/SUNs) within the Munich Urban Carbon Column network (MUCCnet). OCO-2 XCO2 retrievals from larger-swath target observations are compared to average XCO2 retrieved from MUCCnet ground-based sensors within the target viewing area for a number of comparison days. Furthermore, OCO-2 XCO2 retrievals are also compared to ground-based EM27 XCO2 to evaluate its ability to capture XCO2 gradients within its target viewing area. Overall, the paper presents a valuable scientific contribution to Atmospheric Measurement Techniques and to the satellite community in general in evaluating the ability for OCO-2 to capture gradients within its target viewing mode region. I recommend publication after addressing the minor comments noted below.

General comments:

- There are a number of areas within the manuscript that refer to XCO2 correctly in ppm, but then also refer to this quantity as both a concentration and/or a mixing ratio in various places within the text. The unit of ppm is a mole fraction and does not include a volume quantity, so please be cautious and consistent with the unit of XCO2 (ppm) throughout the manuscript.
- In general, the manuscript is very well-written and concise, but there are a number of grammatical errors and typos within the manuscript that could be removed with more thorough editing. I have noted some corrections below in the technical comments.
- Figure 2: The units of XCO2 are missing on subplots. It also looks like the target observations overlap quite a bit – how do you deal with this in the comparison methods and does averaging overlapping soundings affect your additional quality flag results in Figure 3? Similarly, Figure 15 is also missing XCO2 units and overlapping observations make it difficult to compare to the model. Is it possible to average these observations in some way to better visualize this comparison?
- For the model comparison, it wasn't clear within the text whether or not the WRF XCO2 calculation takes into account the OCO-2 averaging kernel. Does it? Without doing so, the comparison between OCO-2 and the model is not a true 1:1 comparison.

Specific comments:

L24: please replace "concentrations" with "mole fractions"

L31: Reference is not valid and is also not included in the References section

L39: It would be worthwhile to state that the TCCON monitors the long-term atmospheric growth of total-column CO2, CO and CH4…

L54: I think that the 14-day repeat cycle is incorrect, isn't it 16 days?

L66: Some care should be taken here to define the scale at which OCO-2 may be able to resolve XCO2 fluxes, which is dependent upon the spatial scale of the target itself.

L106: Because it is not possible to truly calibrate XCO2, it is somewhat misleading to call this quantity XCO2 "calibrated". Rather, this is a bias-corrected and scaled XCO2 retrieval that is presented.

L125: Please state to which CO2 scale these retrievals are tied for the reader.

Figure 3 caption: Should "spectrometer locations […]" be moved to Figure 2?

L138: Given the reference to Figure 2 and this statement, perhaps the order of Figure 2 and Figure 3 should be switched?

L144-155: Does the model XCO2 calculation take into account the OCO-2 averaging kernel?

L172: Can you explain more how the collocation radius is chosen? Is this 6 km chosen to equally segment the target area around the EM27 sensors?

L192: Please include punctuation around equations here and elsewhere.

L226: Caution should be taken in stating that OCO-2 "measures" XCO2 mole fractions. Rather, OCO-2 measures a radiance that is converted to a mole fraction of XCO2 via a retrieval algorithm. In this sense, care should be taken to refer to XCO2 from both ground-based and spaceborne instrumentation as "retrievals" here, in Figure 8, and elsewhere throughout the manuscript.

L247: Other potential causes for differences in biases are explained in L254-257, so perhaps it's worth moving this discussion to this paragraph.

Figure 11: These error bars look very small given the error bars in OCO-2 XCO2 in Figure 9. Can you explain a bit more how you've calculated this error? In addition, given that the gradient in XCO2 is very small, it might be worth indicating minor x-axis grid lines here for the reader.

L299: Given that the gradients presented in Figure 11 do not have overlapping error bars and therefore do not represent, qualitatively, similar mean XCO2 differences, would it be more accurate to state that OCO-2 is capable of *detecting intra-target XCO2 gradients of a similar sign as MUCCnet XCO2*?

L327: Again, here I am wondering how the XCO2 from the WRF model is computed because L144-155 does not describe whether or not this XCO2 quantity derived takes into account the OCO-2 averaging kernel.

L331: 'mixing ratios' are described in this same sentence in addition to mole fractions. Please use mole fractions and maintain consistency throughout the manuscript.

L344: It would be useful either here or in a previous section to describe what "good measurement conditions" entails.

L345: Similar to L66, "middle" to "larger" sized cities might be irrelevant without a spatial scale. It would be helpful to state the spatial scale that you would expect OCO-2 to be able to resolve urban XCO2 gradients, given the swath area of the target.

Technical comments:

L10: Please change "constraint" to "constrained"

L46-47: Consider rephrasing for readability

L93: … determined "by" performing

L153: Please change "weighing" to "weighting"

L173: Please change "none" to "no"

L223: "Observing systems observe" is somewhat redundant.

L240: Please replace "due to the " with "by"

L290: Please replace "shows" with "shown"

---

## Author Comment (AC1)

Dear reviewer 1,
We are very grateful for your comments and suggestions, which have helped to improve our manuscript significantly. We have revised the manuscript accordingly. The following is a point to point response to your comments and suggestions. Corresponding changes in the manuscript are also made available below at the appropriate places, if applicable.

Sincerely,
Maximilian Rißmann and Jia Chen on behalf of all co-authors.
* * *
**There are a number of areas within the manuscript that refer to XCO2 correctly in ppm, but then also refer to this quantity as both a concentration and/or a mixing ratio in various places within the text. The unit of ppm is a mole fraction and does not include a volume quantity, so please be cautious and consistent with the unit of XCO2 (ppm) throughout the manuscript.**
*Thanks for this comment. We indeed have not been too consistent in the use of the term column averaged dry air mole fractions (DMF). We updated the manuscript accordingly.*

**In general, the manuscript is very well-written and concise, but there are a number of grammatical errors and typos within the manuscript that could be removed with more thorough editing. I have noted some corrections below in the technical comments.**
*We incorporated all the technical comments mentioned below. Thanks so much for taking the effort of compiling the list of corrections!*

**Figure 2: The units of XCO2 are missing on subplots. It also looks like the target observations overlap quite a bit – how do you deal with this in the comparison methods and does averaging overlapping soundings affect your additional quality flag results in Figure 3? Similarly, Figure 15 is also missing XCO2 units and overlapping observations make it difficult to compare to the model. Is it possible to average these observations in some way to better visualize this comparison?**
*Thank you very much for this suggestion. We updated all the OCO-2 target figures in our manuscript by averaging OCO-2 soundings into 0.02° x 0.02° bins. XCO$_2$ units are added to the plots. We do not introduce any weighting terms to deal with overlapping soundings in spaceborne XCO$_2$ retrievals. Each good-quality OCO-2 sounding contributes the same to the mean. Hence, the averaging of overlapping soundings also does not affect the additional quality flag results in Figure 3.*

**For the model comparison, it wasn't clear within the text whether or not the WRF XCO2 calculation takes into account the OCO-2 averaging kernel. Does it? Without doing so, the comparison between OCO-2 and the model is not a true 1:1 comparison.**
*Thanks for the suggestion. We performed an averaging kernel correction. Please see our answer in the comment referring lines 144-155.*

**L24: please replace "concentrations" with "mole fractions"**
*Done, thanks!*

**L31: Reference is not valid and is also not included in the References section**
*We updated the reference in the manuscript to "Montzka, S.: The NOAA Annual Greenhous Gas Index (AGGI), NOAA Global Monitoring Website, https://gml.noaa.gov/aggi/aggi.html, Last accessed on 07.12.2021, 2021"*

**L39: It would be worthwhile to state that the TCCON monitors the long-term atmospheric growth of total-column CO2, CO and CH4…**
*Thanks for this correction. We changed the according sentence in the manuscript as followed:*

| L39 | It monitors the long-term atmospheric growth of XCO2, XCO and XCH$_4$ along with other atmospheric trace gases. Regular calibrations against aircraft measurements make the TCCON |

| | stations currently the primary validation source for most space-based XCO₂ data products (GOSAT, GOSAT-2, OCO-3, TROPOMI). |
|---|---|

**L54: I think that the 14-day repeat cycle is incorrect, isn't it 16 days?**
*Correct, thanks. We updated the manuscript.*

| L54 | NASA's Orbiting Carbon Observatory instruments (OCO-2 and OCO-3) capture XCO2 in four different measurements modes: nadir, glint, target and snapshot area mode (SAM). OCO-2 captures XCO2 on a 16-day ground-track repeat cycle. |
|---|---|

**L66: Some care should be taken here to define the scale at which OCO-2 may be able to resolve XCO2 fluxes, which is dependent upon the spatial scale of the target itself.**
*We assume cities larger than Munich, which currently has a population of 1.48 million, to generally have larger $CO_2$ emissions and consequently larger spatial gradients. Thanks to your comment we now emphasize in the manuscript that the spatial scale is limited by the 15x20 km target field size of OCO-2. Due to the small size of Munichs inner city (319.71 square kilometers), OCO-2's target area covers a large part of the city. Complementarily, the bigger OCO-3 SAM field-of-view is better suited for emission studies in larger agglomerated areas. Our paper for the first time covers spatial gradients within one OCO-2 target observation, which is made possible by the proximity of the ground-based MUCCnet spectrometers. The relatively low gradients measured during the overpasses of our study enabled us to test the lower detection limits for intra-urban XCO2 fluxes of OCO-2.*

| L66 | This way, we test the capability of OCO-2 to resolve small-scale urban XCO₂ fluxes in Munich and other cities from space, which is needed to study sector dependent emissions in the future. Due to OCO-2's relatively small target size of around 300 square kilometers, the instrument is best suited for spaceborne emission studies in smaller cities while OCO-3's SAM measurements cover a wider field-of-view, which enables the assessment of large metropolises around the globe. |
|---|---|

**L106: Because it is not possible to truly calibrate XCO2, it is somewhat misleading to call this quantity XCO2 "calibrated". Rather, this is a bias-corrected and scaled XCO2 retrieval that is presented.**
*Appreciated. We updated our manuscript accordingly.*

| L108 | All results of this paper are based on the scaled and bias-corrected retrievals XCO2$_{corrected.}$ |
|---|---|

**L125: Please state to which CO2 scale these retrievals are tied for the reader.**
*The OCO-2 v10 data scaling is part of the OCO-2 bias-correction procedure that is applied to the XCO2 soundings in the OCO-2 lite files. The global scaling factor applied, ties the OCO-2 data to TCCON and consequently to the WMO X2007 trace gas scale.*

**Figure 3 caption: Should "spectrometer locations […]" be moved to Figure 2?**
*Done.*

**L138: Given the reference to Figure 2 and this statement, perhaps the order of Figure 2 and Figure 3 should be switched?**
*Thanks a lot for this suggestion. We are of the same opinion and reversed the order of Figure 2 and Figure 3.*

**L144-155: Does the model XCO2 calculation take into account the OCO-2 averaging kernel?**
*We did not account for the difference in the averaging kernels initially.  Thanks to your suggestion we have added the OCO-2 averaging kernel in the calculation of the model XCO2. We revised the simulation results, by using the OCO-2 column averaging kernel (AK, right) and the a-priori CO2 profile (left) provided by ACOS L2 Lite output.*

[Figure]

*We performed the averaging kernel correction following the method proposed by O'Dell et al. 2012 ([https://amt.copernicus.org/articles/5/99/2012/amt-5-99-2012.pdf](https://amt.copernicus.org/articles/5/99/2012/amt-5-99-2012.pdf)):*

| Eq. 3 | $C_{ak} = AK * C_{raw} + (I - AK) * C_{apriori}$ |
|---|---|

*The  plot below shows the AK-corrected WRF-profile and the difference between the uncorrected and corrected WRF-profile.*

[Figure]

[Figure]

**L172: Can you explain more how the collocation radius is chosen? Is this 6 km chosen to equally segment the target area around the EM27 sensors?**

*Correct, the 6 km radius was chosen, to segment the target area around the EM27 sensors as evenly as possible. It provides the largest number of retrievals within each of the three comparison domains within the target field of view while minimizing the overlap of our comparison domains as stated in lines 179-182.*

| L179 | For a collocation radius of $r_{col} = 6$ km around the spectrometer locations we achieve the highest number of collocated soundings for each site while having almost no overlap of collocated soundings between the sites (most soundings are collocated to only one MUCCnet site). This way, we segment the target observation data into three comparison domains - centre, west and north. |
|------|---|

*A larger comparison set of soundings also reduces the effect of random errors in our computed mean XCO2. We assume this relatively large comparison domain to best represent the actual XCO2 around our ground-based measurement sites. We also added this explanation to the manuscript to further describe our thought process when choosing the comparison domains.*

| L182 | A large comparison set of soundings also reduces the effect of random errors in our computed mean XCO2. We assume this relatively large comparison domain to best represent the actual XCO2 around our ground-based measurement sites. |
|------|---|

**L192: Please include punctuation around equations here and elsewhere.**
*Implemented.*

**L226: Caution should be taken in stating that OCO-2 "measures" XCO2 mole fractions. Rather, OCO-2 measures a radiance that is converted to a mole fraction of XCO2 via a retrieval algorithm. In this sense, care should be taken to refer to XCO2 from both ground-based and spaceborne instrumentation as "retrievals" here, in Figure 8, and elsewhere throughout the manuscript.**
*Again, we want to excuse this oversight. We have modified the statement in Figure 8 and now use retrieve\retrievals throughout the manuscript.*

**L247: Other potential causes for differences in biases are explained in L254-257, so perhaps it's worth moving this discussion to this paragraph.**
*We moved the sentence to the paragraph starting in L254. Thanks for this suggestion. The paragraph now reads as follows:*

| L255 | The daily offsets in each domain are depicted in Fig. 10. We assume, measurement uncertainties and the relatively small sample size of eleven overpass days to cause the discrepancies in the computed mean biases of the three collocation domains. OCO-2 retrieves higher CO2 mole fractions than MUCCnet in all three domains, during each overpass except for August 12,2020. For most overpasses the by-site offsets are consistent in each of the three collocation areas. The largest discrepancies in daily offsets in the three domains could be observed on overpass days with a smaller than average number of good quality soundings (e.g. November 9, 2020 and July 27, 2020) |
|------|---|

**Figure 11: These error bars look very small given the error bars in OCO-2 XCO2 in Figure 9. Can you explain a bit more how you've calculated this error? In addition, given that the gradient in XCO2 is very small, it might be worth indicating minor x-axis grid lines here for the reader.**
*Thanks for this question. We calculated the standard error of the mean and introduced a weighting term (number of samples in the domain) to the error propagation of the standard because we wanted to set the focus on the robustness of the mean. Here we need to address, that we took a really simple approach to the error in the mean $X_{CO2}$ in the comparison domain. As another reviewer pointed out, we made the simplifying assumption, that the errors of XCO2 soundings to be randomly distributed within our comparison domain, which is not the case. We*

*neglected systematic errors in the OCO-2 retrievals. To maintain consistency in our results (consistency with Figures 7 and 8) we removed the weighing term when computing the error for the XCO2 gradient assessment, to compute the combined standard deviation:*

| Eq. 6 | $sd_{domain1-domain2} = sqrt(\sigma_{domain1}{}^2 + \sigma_{domain2}{}^2).$ |
|---|---|

*We revised our representation of errors in the gradients and updated the plot accordingly.*
*This representation better shows the variance of single soundings within the comparison domains especially for the spaceborne retrievals. Even though the standard deviation is relatively large on some days, the overall RMSD of captured spaceborne gradients when compared to MUCCnet is 0.31 ppm, which indicates the instruments capability of resolving spaceborne gradients on a sub-city scale. We updated figure 11 as shown below.*

[Figure]

**Figure 11.** XCO2 gradients in Munich on overpass days. Blue bars represent the gradients present in the OCO-2 target observations. Orange bars denote XCO2 gradients captured by MUCCnet. On most days, OCO-2 sees elevated XCO2 in the same region as the ground-based MUCCnet instruments. Error bars are computed using the combined standard deviations of the XCO2 samples in the two domains which are used to compute gradients. (see Eq. 6).

**L299: Given that the gradients presented in Figure 11 do not have overlapping error bars and therefore, do not represent, qualitatively, similar mean XCO2 differences, would it be more accurate to state that OCO-2 is capable of detecting intra-target XCO2 gradients of a similar sign as MUCCnet XCO2?**
*You are correct. The non-overlapping error bars were caused by our previous, non-ideal representation error in the computed gradients. Normalizing the error in each domain with the number of soundings in that domain was not a realistic representation of the actual variance of the computed mean gradients.*
*We also noticed OCO-2's strength of replicating the actual sign (or direction) of the ground-based gradients, which we stated in two paragraphs of the manuscript.*

| L277 | Considering the rather small XCO2 gradients in Munich, OCO-2 detects the elevated XCO2 in the same domain as MUCCnet for 20 of the 22 computed gradients and therefore qualitatively determines the area of enhanced XCO2 correctly in 91% of cases. |
|---|---|
| Figure 11 | On most days, OCO-2 sees elevated XCO2 in the same region as the ground-based MUCCnet instruments. |

*Nonetheless, the overall RMSD of captured spaceborne gradients when compared to MUCCnet is around 0.31 ppm, which indicates the instrument's capability of capturing similar mean XCO2 gradients on a sub-city scale. This is also stated in the manuscript in line 291:*

| L291 | For the entire set of gradients OCO-2 achieved an RMSD of 0.31 ppm and a linear correlation with a strong correlation of $R^2$= 0.68 between OCO-2 and the MUCCnet measurements |
|---|---|

**L327: Again, here I am wondering how the XCO2 from the WRF model is computed because L144-155 does not describe whether or not this XCO2 quantity derived takes into account the OCO-2 averaging kernel.**
*We take the averaging kernels into account now. See our answers to your comments above.*

**L331: 'mixing ratios' are described in this same sentence in addition to mole fractions. Please use mole fractions and maintain consistency throughout the manuscript.**
*Done.*

**L344: It would be useful either here or in a previous section to describe what "good measurement conditions" entails.**
*We updated the sentence. Generally, our filter criteria are determined by the number of good-quality soundings OCO-2 was able to retrieve in each comparison domain. The number of good quality soundings is mostly determined by the cloud coverage during the overpass time. In this study we only consider overpass days on which OCO-2 retrieved a minimum of 500 good quality soundings. We cover the selection criteria for the gradient comparison in more details in Section 3.3 – Gradient Comparison.*

| L344 | *These results suggest that for high gradients and cloud free measurement conditions, OCO-2 target observations can be utilized for an accurate assessment of urban XCO2 and its spatial distribution.* |
|---|---|

.

**L345: Similar to L66, "middle" to "larger" sized cities might be irrelevant without a spatial scale. It would be helpful to state the spatial scale that you would expect OCO-2 to be able to resolve urban XCO2 gradients, given the swath area of the target.**
*This statement was also made under the assumption, that in "larger" more populated areas there are higher XCO2 emissions, that cause greater spatial XCO2 gradients during a measurement. It's right, that the study domain will always be constrained by OCO-2's limited target area. As mentioned above, larger areas then could be covered using OCO-3 SAM observations. We removed the vague statement from the manuscript.*

**Technical comments:**
**L10: Please change "constraint" to "constrained"**

| L10 | Due to this more constrained collocation, we observe improved agreement between space-borne and ground-based XCO2 in all three comparison domains. |
|---|---|

**L46-47: Consider rephrasing for readability**

| L46 | In recent years, EM27/SUN instruments have been used in measurement campaigns that aim to quantify urban anthropogenic emissions by combining differential column measurements (DCM) and atmospheric transport models. Multiple field campaigns have been carried out in Berlin, Munich, Indianapolis, San Francisco, California, Poland, St. Petersburg and Hamburg. These studies show the potential of top-down emission estimates as they can help uncover unknown emission sources and constrain bottom-up emission inventories. |
|---|---|

**L93: … determined "by" performing**

| L93 | The sensors are calibrated by subtracting constant offsets which are determined in side-by-side measurements. |
|---|---|

**L153: Please change "weighing" to "weighting"**

| L153 | XCO2 in the study area is derived from the modelled concentration profiles with an appropriate pressure weighting, following the method described in Zhao 2019. |
|---|---|

**L173: Please change "none" to "no"**

| L173 | For a collocation radius of $r_{col}$=6 km around the spectrometer locations we achieve the highest number of collocated soundings for each site while having almost no overlap of collocated soundings between the sites (most soundings are collocated to only one MUCCnet site). |
|---|---|

**L223: "Observing systems observe" is somewhat redundant.**

| L223 | Both observing systems capture a similar seasonal behaviour of urban XCO2 in Munich in the time period analyzed here |
|---|---|

**L240: Please replace "due to the " with "by"**

| L240 | This improvement is caused by more specific collocation, that reduces the effect of averaging over potential spatial XCO2 gradients in the OCO-2 target observation. |
|---|---|

**L290: Please replace "shows" with "shown"**

| L290 | These improved results are shown … |
|---|---|

---

## Author Comment (AC2)

Dear Chris,

We are very grateful for your comments and suggestions, which have helped to improve our manuscript significantly. We have revised the manuscript accordingly. The following is a point to point response to your comments and suggestions. Corresponding changes in the manuscript are also made available below at the appropriate places, if applicable.

Sincerely,
Maximilian Rißmann and Jia Chen on behalf of all co-authors.
* * *
**The first major comment involves error bars. The error bars plotted on OCO2 biases (relative to MUCCnet), such as shown in Figures 7 & 8, appear to be standard deviation only. \*IF\* one were to assume that these errors were randomly distributed over the small area (<10 km) over which they are evaluated, we could estimate the standard error of the mean as sigma/sqrt(N), where N is the number of observations. However, it is well known that OCO-2 errors are NOT randomly distributed, over small or large areas (see e.g. Kulawik et al., 2019). It is likely that the local scale mean of OCO-2 includes systematic errors that cannot easily be evaluated. This point should clearly be made. Therefore, it is difficult to evaluate errors on mean OCO-2 values over the whole domain or the 3 sub-domains. Further, it is equally difficult to state them for the gradients. On this note, the error treatment on the gradients is both incorrect (as it assumes 1/sqrt(N) Gaussian averaging) and inconsistent with the stated error bars on XCO2 overpass means (which simply uses the standard deviation of the XCO2 values in a given domain). The current error bars listed on the gradients are unrealistically small (of order or less than 0.1 ppm). Therefore, please expand your discussion of errors to include these points, and be sure to treat errors consistent on the mean quantities and the gradient quantities.**

*Thank you very much for pointing this out. We indeed approached the errors of the computed XCO2 mean in a very simplistic way. We made the simplifying assumption, that the errors of XCO2 gradients are randomly distributed within our comparison domains, which is not the case. We neglected systematic errors in the OCO-2 retrievals. To maintain consistency in our results we removed the weighing term when computing the error of the $X_{CO2}$ gradients, to compute the combined standard deviation. We revised our representation of errors in the gradients and updated the plot accordingly to be consistent with the standard deviation of the $X_{CO2}$ samples we show in Sections 4.1 & 4.2 (Figure 7 & 8).:*

[Figure]

| Eq. 6 | $sd_{domain1-domain2} = sqrt(\sigma_{domain1}{}^2 + \sigma_{domain2}{}^2)$ |
| --- | --- |

*We updated Figure 11 accordingly:*

[Figure]

**Figure 11.** XCO2 gradients in Munich on overpass days. Blue bars represent the gradients present in the OCO-2 target observations. Orange bars denote XCO2 gradients captured by MUCCnet. On most days, OCO-2 sees elevated XCO2 in the same region as the ground-based MUCCnet instruments. Error bars are computed using the combined standard deviations of the XCO2 samples in the two domains which are used to compute gradients. (see Eq. 6).

*We further added a short paragraph to emphasize, that we are not representing a fully correct description of the error in the mean XCO2 gradients.*

| L215 | Rather than the error of the mean we represent the combined spread of XCO2 in the two domains. |

*In case there is a better way of representing the error in the XCO2 gradients, we would greatly appreciate further suggestions and are happy to incorporate them into our manuscript.*

**Second comment (albeit minor) is how the data are presented on a map. Figure 2 shows this for the 12 dates analyzed. I suggest averaging the data onto a 0.5x0.5 km2 grid, because as you show it now, it is large circles that overlap each other. Your plotting method emphasizes the noise in the data, rather than the spatial gradients. It further ignores that rather large horizontal extent of the OCO-2 FOVs, which are parallelograms and in some of these cases are rather wide! You can see an example of this in the figure below, which shows your plot of the Dec 18, 2020 case on the left, the corresponding NASA Worldview plot in the middle (obtainable at https://go.nasa.gov/38MQLRr), and my own home-made generated plot on the right. You can see the two on the right de-emphasize the noise and emphasize the gradients, and further do a better job illustrating the fact that OCO-2 target observations provide relatively complete spatial coverage due to the large degree of overlap of the various soundings in the image. The central image also illustrates the value of including the actual Aqua-MODIS data, at the expense of the spatial resolution of the surface imagery (unfortunately). But often there is cloud contamination in these Munich targets, so it is something to consider when plotting. You may wish to include something like the central image in your Figure 15, to really showcase the cloud context. And in general, please consider using one of these more realistic plotting methods which includes the actual parallelogram-shaped along with some kind of averaging.**

*We appreciate this suggestion. We updated all the OCO-2 target figures in our manuscript by averaging OCO-2 soundings into 0.02°x 0.02° km bins to comply with the standard guidelines given by JPL. Furthermore, we added the XCO$_2$ units to the map-plots.*

[Figure]

**Specific comments:**

Section 3.2: The value of 6km radius around each EM27 seems rather arbitrary. Yes, it maximizes the circles while minimizing their overlap, but does it make sense as to the column of air the various MUCCnet sites are really sampling, as compared to OCO-2? For instance, using +/- 30 minutes overpass time for collocation, we could estimate that corresponds to a spatial extent of roughly +/- 15 km for a wind speed of 8 m/s, meaning

**that the EM27 signals will be much more "washed out" in the case of higher wind speeds, due to averaging all the EM27 values in that 1 hour time window. You should at least speak to this source of collocation error, and you may wish to mention that more sophisticated schemes (such as using a tighter time window) may reduce the collocation error.**

*Thank you very much for highlighting this issue. We chose the 6 km radius to equally segment the target area as evenly as possible around MUCCnet's ground-based measurement sites. According to the ERA-5 v10 wind data we used, windspeeds in Munich are rather low. The average wind speed over all was 2.33 +/- 1.54 m/s which corresponds to a spatial extend of around 4.5 km within our colocation time. The highest average windspeeds of 5.08 +/- 1.60 m/s are provided on February 13, 2021. Due to the low wind speeds, we assume the EM27/SUNcolumn signals to still coincide with the spatial extend of our comparison domains. In addition, we usually see rather low temporal variances within the ground-based XCO2 retrievals when compared to the larger variance in the collocated spaceborne XCO2. Consequently, we trade a larger number of spaceborne soundings to reduce the impact of random errors in single soundings against higher specificity in the EM27 signals.*

*We added a short paragraph in to clarify our thought process:*

| L194 | The relatively long collocation time frame is chosen due to the low average wind speeds of 2.33 +/- 1.54 m/s during the overpasses featured in this study. This may, especially for higher wind speeds, introduce collocation error which can be reduced by adjusting the collocation time frame according to the wind speed. |
|---|---|

**Near line 150: Please state the spatial resolution of the anthropogenic emissions in TNO-GHG? It appears to be roughly 1x1 km^2, but if it is larger, than could also explain the Dec 18, 2020 discrepancy.**

Near surface emission fluxes are taken from the first version of the GHG and co-emitted species emission database provided by the Netherlands Organization for Applied Scientific Research for 2015 (TNO-GHGco_v1.1). The innermost domain covers Munich and its surrounding, initialized by a higher-resolution version of TNO_GHGco_v1.1 at a resolution of approx. 1 km × 1 km. Details of the modelling setup can be found in Zhao et al., (2022). (https://acp.copernicus.org/preprints/acp-2022-281/acp-2022-281.pdf).

**Near line 335: Please state the WRF-GHG wind speeds (roughly) in this case. If they are significantly larger than the 0.5-2 m/s wind speeds ERA5 shows, that could explain the ~factor of 10 discrepancy.**

*WRF modelled mean wind speeds at the 10 meters above the ground within ± 2h of the overpass time are around 1.44 m/s with its standard deviation of 0.48 m/s. Unfortunately, we do not have a better knowledge about the actual windspeed and direction during the overpass, since we did not run any measurement. Nonetheless, we assume the relatively large discrepancies between the modelled XCO2 and the OCO-2 XCO2 to be caused by one or more of the following reasons:*

*First of all, December 18th, 2020 was an outlier in terms of the captured XCO2 gradients. The gradients are considerably larger, that during all other overpasses. This is true for gradients captured by MUCCnet as well as OCO-2. Since we compare to XCO2 data, that was generated by a yearly averaged emission inventory, we do not expect the model to reliably replicate the actual gradients on that day. In addition, the modeled plume of the WRF-GHG simulation is shifted northwards when compared to the OCO-2 target observation. This causes the highest XCO2 of the simulation to not be fully sampled by the western collocation domain resulting in lower enhancements between the two domains. In the center of the plume, WRF-GHG models XCO2 enhancements of around 0.7 ppm, which is closer to the gradients captured by MUCCnet and OCO-2. We also state this similarly in the manuscript:*

| L350 | We assume this underestimation of XCO2 gradients to be caused by both, uncertainties in the annual emission inventory as well as transport uncertainties. A mismatch in model wind speed and direction causes the area of maximum XCO2 enhancements to be shifted to the north in the modeled XCO2 (see Fig. 15). Furthermore, the XCO2 in the target observation is notably higher than on other days, indicating unusually high emissions in Munich on December 18, 2020, which can't be replicated by a yearly averaged bottom-up emission inventory, while the spatial distribution is reproduced rather accurately. |
|---|---|

**Near line 365: You ignore the factor of 10 discrepancy in the spatial XCO2 gradients between WRF-GHG and the observations. You should mention this as well, as the obs provide a potential way to improve whatever is going on in the model. I'm guessing perhaps there is something fundamentally wrong about how the model was set up and run, because I think WRF, at sufficiently high spatial resolution, should be able to duplicate the rough magnitude of observed spatial gradients**

*As stated above we think this factor of 10 is rather caused by the way we compute the gradients in this case. We compare the mean XCO2 of the center domain to the mean XCO2 in the western domain. Due to the shifted plume in the WRF-GHG model, we do not sample the actual highest XCO2 in the model data, while we do that for the OCO-2 measurements.*

---

## Author Comment (AC3)

Dear reviewer 3,

We are very grateful for your comments and suggestions, which have helped to improve our manuscript significantly. We have revised the manuscript accordingly. The following is a point to point response to your comments and suggestions. Corresponding changes in the manuscript are also made available below at the appropriate places, if applicable.

Sincerely,
Maximilian Rißmann and Jia Chen on behalf of all co-authors.
* * *
**Colocation criteria: I have some concerns about the time selection of +/- 30 min. It seems to be too long especially when looking at gradient signals between closely located sites (< 10 km). Rather there should be a good compromise between the colocation geometry and the time. To motivate the choice made by the authors a sensitivity study for one selected day (with highest wind speed) would help.**

*We appreciate this comment. We had a few reasons for the selection of the collocation time. First of all ,a constant collocation time of +/- 30 mins makes our work better comparable to other studies that compared and validated OCO-2 against ground based measurements. Second we believe, that due to the relatively low average wind speeds of 2.33 +/- 1.54 m/s during the overpasses featured in our study, the collocation time of +/- 30 mins is justified. Due to these low wind speeds, we assume the EM27/SUN column signals to coincide with the spatial extend of our comparison domains during the entire comparison time frame. Changing the collocation time also had relatively small effects on the overall mean XCO2 measured by the MUCCnet spectrometers. Consequently, we do not think that for our study the colocation time was too long.*

*We added a short paragraph about the wind speeds to our manuscript.*

| L192 | The relatively long collocation time frame is chosen due to the low average wind speeds of 2.33 +/- 1.54 m/s during the overpasses featured in this study. |
|---|---|

**L88: Reference Frey and Gisi, 2021 is not correctly labeled / missing doi or link?**

*Here we refer to the official PROFFAST calibration guidelines for EM27 instruments of the COCCON network. Hence, there is no doi. We updated the reference in the script.*

| L89 | This indirectly ties the MUCCnet XCO2 retrievals to the TCCON site in Karlsruhe since the COCCON reference device is calibrated against the TCCON site in Karlsruhe (Alberti et al., 2022; Frey and Gisi, 2021). |
|---|---|
| L407 | Alberti, C., Hase, F., Frey, M., Dubravica, D., Blumenstock, T., Dehn, A., Castracane, P., Surawicz, G., Harig, R., Baier, B. C., Bès, C., Bi,J., Boesch, H., Butz, A., Cai, Z., Chen, J., Crowell, S. M., Deutscher, N. M., Ene, D., Franklin, J. E., García, O., Griffith, D., Grouiez, B.,Grutter, M., Hamdouni, A., Houweling, S., Humpage, N., Jacobs, N., Jeong, S., Joly, L., Jones, N. B., Jouglet, D., Kivi, R., Kleinschek, R.,Lopez, M., Medeiros, D. J., Morino, I., Mostafavipak, N., Müller, A., Ohyama, H., Palmer, P. I., Pathakoti, M., Pollard, D. F., Raffalski, U.,410Ramonet, M., Ramsay, R., Sha, M. K., Shiomi, K., Simpson, W., Stremme, W., Sun, Y., Tanimoto, H., Té, Y., Tsidu, G. M., Velazco, V. A.,Vogel, F., Watanabe, M., Wei, C., Wunch, D., Yamasoe, M., Zhang, L., and Orphal, J.: Improved calibration procedures for the EM27/SUN spectrometers of the COllaborative Carbon Column Observing Network (COCCON), Atmospheric Measurement Techniques, 15, 2433–2463, https://doi.org/10.5194/amt-15-2433-2022, 2022. |
| L427 | Frey, M. and Gisi, M.: Calibration of the EM27 / SUN Instruments, https://www.imk-asf.kit.edu/downloads/Coccon/2021-04-30_Instrument-Calibration.pdf, 2021 |

**L131: should be "starting from April, 2020"**

*Done, thanks for noticing!*

---

## Referee Report (RR1)

The updated version of "Comparison of OCO-2 target observations to MUCCnet - Is it possible to capture urban $X_{CO2}$ gradients from space?" by Riessmann, et al., is much improved over the original (already quite good) manuscript. After addressing the two minor comments below, it will be ready for publication.

**Figure 3 Caption** must describe the 0.02x0.02 degree gridding scheme that is used to present the OCO-2 data.

**Lines 154-158, regarding the AK Correction**. "AK" in your equation (3) appears to be the 20x20 CO2 averaging kernel matrix, which is not available in the lite files. It appears you used the AK term for XCO2 (a vector, not a matrix, and which I typically call "a"), which if you did it that way is actually wrong. That this is a matrix is clearly stated in O'Dell et al. (2012) right after the equation is introduced. Rather than using the "C" profile quantities, you may wish to put the equation in XCO2 form for simplicity, and which does not require the full AK matrix. The proper AK equation in XCO2 form is as follows, and actually can be written three different ways:

XCO2_ak = Sum{i=1..nlev}  h_i a_i u_mod,i + h_i (1-a_i) u_ap,i + [ h_i u_mod,i – h_i, u_mod,i]
            = XCO2_mod + Sum{i=1..nlev} h_i (1-a_i) (u_ap,i – u_mod,i)
            = XCO2_ap + Sum{i=1..nlev} h_i a_i (u_mod,i – u_ap,i)

where "u" are profile co2 mole fractions, "ap" means the retrieve a priori, "mod" means model, "h" is the pressure weighting function vector, and "a" is the normalized averaging kernel vector for XCO2 (with values typically ranging from 0 – 1.5) . Both forms of the equation are easily derived from your equation (3), where "AK" from equation (3) is the full averaging kernel *matrix* (not vector!). "h" and "a" are both quantities available in the OCO-2 lite files. The middle form of the AK correction equation above shows that the correction only modifies the "pressure-weighted" XCO2 from the model if some levels have "a" significantly different than unity, and the apriori value is significantly different from the model value for those same levels. The third form of the equation was given in Connor et al. (2008, https://agupubs.onlinelibrary.wiley.com/doi/full/10.1029/2006JD008336), as his equation (15).

---

## Author Response (AR2)

Dear Chris,

We are very grateful for your comments and suggestions, which have helped to improve our manuscript significantly. We have revised the manuscript accordingly. In the following we respond to your two comments made on the second revision.

Sincerely,
Maximilian Rißmann and Jia Chen on behalf of all co-authors
* * *
**Figure 3 Caption must describe the 0.02x0.02 degree gridding scheme that is used to present the OCO-2 data.**

*Thanks a lot for pointing this out. We adjusted the Figure caption accordingly.*

**Lines 154-158, regarding the AK Correction. "AK" in your equation (3) appears to be the 20x20 CO2 averaging kernel matrix, which is not available in the lite files. It appears you used the AK term for XCO2 (a vector, not a matrix, and which I typically call "a"), which if you did it that way is actually wrong. That this is a matrix is clearly stated in O'Dell et al. (2012) right after the equation is introduced. Rather than using the "C" profile quantities, you may wish to put the equation in XCO2 form for simplicity, and which does not require the full AK matrix. The proper AK equation in XCO2 form is as follows, and actually can be written three different ways:**
**XCO2_ak = Sum{i=1..nlev} h_i a_i u_mod,i + h_i (1-a_i) u_ap,i + [ h_i u_mod,i – h_i, u_mod,i]**
**= XCO2_mod + Sum{i=1..nlev} h_i (1-a_i) (u_ap,i – u_mod,i)**
**= XCO2_ap + Sum{i=1..nlev} h_i a_i (u_mod,i – u_ap,i)**
**where "u" are profile co2 mole fractions, "ap" means the retrieve a priori, "mod" means model, "h" is the pressure weighting function vector, and "a" is the normalized averaging kernel vector for XCO2 (with values typically ranging from 0 – 1.5) . Both forms of the equation are easily derived from your equation (3), where "AK" from equation (3) is the full averaging kernel matrix (not vector!). "h" and "a" are both quantities available in the OCO-2 lite files. The middle form of the AK correction equation above shows that the correction only modifies the "pressure-weighted" XCO2 from the model if some levels have "a" significantly different than unity, and the apriori value is significantly different from the model value for those same levels. The third form of the equation was given in Connor et al. (2008, https://agupubs.onlinelibrary.wiley.com/doi/full/10.1029/2006JD008336), as his equation (15).**

*Thanks for your comment regarding the Eq.3 and sorry for this misunderstanding. You are right, the lite file does not contain the 20 x 20 matrix, but rather the averaging kernel vector. In this study, we use the mean of these profiles over the target area around Munich to smooth the WRF modelled concentration profiles, which is shown in Fig.1 (on the next page). Next, we indeed follow the method presented in O Dell et al. (2012) to calculate the modelled AK-smoothed concentration $XCO_{2,ak}$:*

$$XCO_{2,ak} = \sum_{i=1}^{n_{lev}} h_i[a_i u_{mod,i} + (1 - a_i)u_{ap,i}]$$

*Here, $a_i$ and $h_i$ denote the AK value and pressure weight at the $i^{th}$ model level, $u_{mod,i}$ and $u_{ap,i}$ represent the modelled and a-priori CO2 concentrations at the $i^{th}$ model level.*

*We corrected Equation 3 in the manuscript to now be consistent with our method:*

| Eq. 3 | $$XCO_{2,ak} = \sum_{i=1}^{n_{lev}} h_i[a_i u_{mod,i} + (1 - a_i)u_{ap,i}]$$ |
|-------|-------|

[Figure]

*Figure 1. The AK profiles over the area of our interest around Munich (grey) and their mean used in the calculation of smoothing the modelled concentration (red).*

---

## Author Response (AR3)

Dear Joanna,

We are very grateful for your comments and suggestions, which have helped to improve our manuscript significantly. We have revised the manuscript accordingly. In the following we respond to your comments. Our changes are also highlighted in the track changes file.

Sincerely,
Maximilian Rißmann and Jia Chen on behalf of all co-authors
* * *
**I have only a couple of technical suggestions. In the Figure 7 caption it would be good to specify that this comparison is for the center instrument and includes all good soundings within the target area. Perhaps the smaller domains for comparisons can also be mentioned in the Figure 9 caption. This would help the reader who is doing figure browsing.**

Thank you for noticing this. We adjusted the Figure captions accordingly.

**On lines 235 and 236, I didn't understand the difference between seasonal behavior and seasonal cycle. Perhaps you could clarify.**

Thank you very much for picking this up. The second sentence indeed was redundant. Thus, we removed the second sentence in line 236 from the manuscript.